# Temporospatial variability of snow's thermal conductivity on Arctic sea ice

**Amy R. Macfarlane**[1]**, Henning Löwe**[1]**, Lucille Gimenes**[1]**, David N. Wagner**[1,2]**, Ruzica Dadic**[1,3]**, Rafael Ottersberg**[1]**, Stefan Hämmerle**[4]**, and Martin Schneebeli**[1]

[1]WSL Institute for Snow and Avalanche Research SLF, Davos Dorf, Switzerland
[2]CRYOS, School of Architecture, Civil and Environmental Engineering, EPFL, Lausanne, Switzerland
[3]University of Wellington, Wellington, New Zealand
[4]SCANCO Medical AG, Bassersdorf, Switzerland

**Correspondence:** Amy R. Macfarlane (amy.macfarlane@slf.ch) and Henning Löwe (loewe@slf.ch)

**Abstract.** Snow significantly impacts the seasonal growth of Arctic sea ice due to its thermally insulating properties. Various measurements and parameterizations of thermal properties exist, but an assessment of the entire seasonal evolution of thermal conductivity and snow resistance is hitherto lacking. Using the comprehensive snow dataset from the Multidisciplinary Drifting Observatory for the Study of Arctic Climate (MOSAiC) expedition, we have evaluated for the first time the seasonal evolution of the snow's and denser snow-ice interface layers' thermal conductivity above different ice ages (refrozen leads, first-year ice, and second-year ice) and topographic features (ridges). Our dataset has a density range of snow and ice between 50 and 900 kg m$^{-3}$, and corresponding anisotropy measurements, meaning we can test the current parameterizations of thermal conductivity for this snow density range. Combining different measurement parameterizations and assessing the robustness against spatial heterogeneity, we found the average thermal conductivity of snow ($< 550$ kg m$^{-3}$) on sea ice remains approximately constant ($0.26 \pm 0.05$ W K$^{-1}$ m$^{-1}$) over time irrespective of underlying ice type, with substantial spatial and vertical variability. Due to this consistency, we can state that the thermal resistance is mainly influenced by snow height, resulting in a 2.7 times higher average thermal resistance on ridges (1411 m$^2$ K W$^{-1}$) compared to first-year level ice (515 m$^2$ K W$^{-1}$). Our findings explain how the scatter of thermal conductivity values directly results from structural properties. Now, the only step is to find a quick method to measure snow anisotropy in the field. Suggestions to do this are listed in the discussion.

## 1 Introduction

Snow's thermal conductivity and insulating properties directly impact heat transfer from the underlying sea ice to the atmosphere and directly inhibit ice growth in the winter season (Sturm and Massom, 2017). Due to this, snow accumulation and snow stratigraphy in winter directly influence the mass balance and, consequentially, the energy balance of sea ice (Eicken et al., 1995; Fichefet and Maqueda, 1997; Sturm et al., 2002a). Snow's thermal conductivity variation stems from the texture, e.g., specific surface area, anisotropy, connectivity, and density (Mellor, 1977; Sturm et al., 1997, 2002a). Understanding this relationship and heterogeneity requires detailed and numerous microstructural snow measurements. The lack of these on Arctic sea ice, due to the inaccessibility of this area in the winter season and shortfalls in the methods (Riche and Schneebeli, 2013), has limited research on the spatial and temporal variability of heat transfer through the snow. Calonne et al. (2019) highlight that the thermal conductivity of snow has previously been widely investigated, whereas studies on firn and porous ice are very scarce. Consequentially, accurately calculating the energy balance variability of sea ice in the high Arctic has considerable shortcomings (West et al., 2020), as we now know that the snow stratigraphy in this region is a com-

plex piece of the puzzle (King et al., 2020; Kaltenborn et al., 2023).

Snow depth and microstructural properties on sea ice are spatially heterogeneous on the meter scale, meaning that heat transfer through the snow cover is highly variable. There are three potential processes of heat transfer through the snow: (1) conduction through the ice; (2) conduction, convection, and radiation across air spaces; and (3) phase change and vapor diffusion between the snow grains (Yen et al., 1991). Conduction and radiation heat transfer through the air spaces is negligible (Sturm et al., 2002a) compared to the conduction of heat through the ice due to the high thermal conductivity of ice. Convection and vapor diffusion depend on the permeability and hence the ice volume fraction of the snow. Due to this, the high ice volume fraction of snow wind slabs on sea ice reduces convection and vapor diffusion. As a result, conduction through the ice is the foremost process influencing heat transfer through the snow cover.

Measuring heat transfer currently has numerous approaches. In the field, the needle probe and heat plate are two destructive but inexpensive methods. Sturm et al. (2002b) were the first and only existing study to measure the thermal conductivity of snow on sea ice directly in the field using a needle probe. The obtained values ranged from $0.078 \, \mathrm{W \, m^{-1} \, K^{-1}}$ for new snow to $0.290 \, \mathrm{W \, m^{-1} \, K^{-1}}$ for an ubiquitous wind slab. This study found a large underestimation when it comes to the average thermal conductivity of snow in comparison to the values inferred from ice growth and temperature gradients ($0.33 \, \mathrm{W \, m^{-1} \, K^{-1}}$). The explanation given for this underestimation was that there was lateral heat transfer within the snowpack, which is not in the $z$ axis. In addition, Riche and Schneebeli (2010) and Fourteau et al. (2022) showed that there were microstructural changes around a needle probe, and measurements do not always reach the required logarithmic regime; these could all be additional reasons for Sturm's underestimation. Lecomte et al. (2013) worked on a density function of sea ice age and thickness whilst referencing Nicolaus et al. (2009), who showed a difference in thermal conductivities on different ice types. Lecomte et al. (2013) concluded that an average thermal conductivity of $0.31 \, \mathrm{W \, m^{-1} \, K^{-1}}$ (Abels, 1892) was too high for snow with an average density of $330 \, \mathrm{kg \, m^{-3}}$. Thermistor strings are in situ measurements that install temperature sensors vertically in the snow and ice (Huwald et al., 2005; Pringle et al., 2007; Marchenko et al., 2019). Thermistor strings measure a continuous time series of temperature gradients within the snow and ice and, in combination with snow thickness data, can be used to compute heat flux through the snow (Sturm et al., 2002b). Unless using an array setup (Pringle et al., 2007), this instrument does not measure spatial variability.

Density is used to parameterize thermal conductivity because of the first-order dependency between thermal conductivity and density. Lecomte et al. (2013) have tested existing parameterizations on their density datasets. It is also a simple, low cost, and quick measurement in the field (Orvig, 1970; Yen, 1981; Fukusako, 1990; Radionov et al., 1997; Sturm et al., 1997; Warren et al., 1999; Sturm et al., 2002a; Domine et al., 2011; King et al., 2020; Arndt, 2022). However, we are now aware of shortcomings when excluding other necessary textural properties from thermal conductivity parameterizations. Developments in X-ray microcomputed tomography ($\mu$CT) techniques have enabled snow research to advance by measuring the exact ice structure without damaging it (Coleou et al., 2001; Riche and Schneebeli, 2010), which allows calculations of the snow density in parallel to the microstructure's textural properties. The microstructure-based finite element method (FEM) of heat conduction through the ice and the air (Arns et al., 2001; Kaempfer et al., 2005; Petrasch et al., 2008; Calonne et al., 2011; Gouttevin et al., 2018) is currently the most reliable method to calculate the thermal conductivity of snow (Riche and Schneebeli, 2013). This opens new opportunities to investigate the relationship between textural properties and heat transport. This method has never been used to measure the thermal conductivity of snow on sea ice.

Löwe et al. (2013) highlight that the sample's anisotropy plays a significant role in the heat transfer through the snowpack and presents a parameterization for thermal conductivity using density and anisotropy for snow, specifically for densities below $500 \, \mathrm{kg \, m^{-3}}$. However, this parameterization is not adapted to high snow densities. Pitman and Zuckerman (1967), Fukusako (1990), Singh (1999), Smith and Jamieson (2014), and Calonne et al. (2019) realized the influence of temperature on the thermal conductivity. Calonne et al. (2019) created upper bounds to ensure that the thermal conductivity is in agreement with the thermal conductivity of ice at specific temperatures in the higher density ranges. However, their parameterization does not include anisotropy. In summary, no current thermal conductivity parameterization includes anisotropy, is precise for high-density snow, and has been tested on snow in the high Arctic.

Given the importance of snow in the sea ice system, we work towards advancing our understanding of both spatial and temporal heterogeneity of the thermal conductivity of snow on sea ice in the high Arctic. We present two new parameterizations, with and without anisotropy, for the complete range of possible snow, firn, and ice densities, developed using microstructure-based FEM using snow samples collected during the Multidisciplinary Drifting Observatory for the Study of Arctic Climate (MOSAiC) expedition (Sect. 2.1). The study of the spatial heterogeneity of the snow on sea ice requires a very high number of measurements, which cannot only be realized by $\mu$CT. A faster method is needed (the $\mu$CT on MOSAiC took 7 h to measure 10 cm of snow). For this reason, we used high-resolution penetrometry using a SnowMicroPen (SMP) to improve the spatial coverage (related individual point measurements to a larger area by increasing the sample size) of individual $\mu$CT profiles (Sect. 2.2) by using SMP density profiles (Sect. 2.3)

to identify both spatial and temporal trends in the dataset (Sect. 3.3). Our measurement concept considered the spatial heterogeneity of sea ice (Macfarlane et al., 2023b). As a result, we can draw new conclusions about the thermal conductivity and resistance of the snow cover on different ice types over the entire winter. This is relevant for calculating the Arctic sea ice's energy budget (Arduini et al., 2022) and allows us to better understand sea ice growth in the winter. Typically, sea ice models use a single layer for the snow cover and a single thermal conductivity and density value (Merkouriadi et al., 2017; Hunke et al., 2017). We compare our dataset to the average snow thermal conductivity value of 0.31 to 0.33 W K$^{-1}$ m$^{-1}$ used in the modeling community (Sturm et al., 2002a; Lecomte et al., 2013; Holland et al., 2021).

## 2 Data and methods

### 2.1 MOSAiC expedition

The field measurements used in this study were conducted during the MOSAiC expedition in the winter months from November 2019 to April 2020 (Nicolaus et al., 2022). The field measurements were located on drifting Arctic sea ice, with the first measurement at 86.3° N, 123.0° E reaching a maximum latitude of 88.9° N and then drifting south until 83.7° N, 13.0° E. A single ice floe was studied in this period. We set up snowpit sites on the sea ice to understand the snow conditions, where we took weekly measurements. These were marked with flags so we could relocate the same snowpit site on the next visit and create a time series of measurements at that location. The snowpit sites were randomly distributed across the ice floe to sample different ice types (e.g., first-year ice (FYI), second-year ice (SYI), and refrozen leads) and topographic features (e.g., ridges). However, the exact location cannot be sampled twice due to the destructive nature of most measurements within the snowpit. The snowpit operator measured consecutive snowpits approximately 1 m apart to continue a time series at one snowpit site. Locations of each snowpit site are indicated alongside the dataset (Macfarlane et al., 2021a). A snowpit is a collection of measurements measuring the physical properties of the snowpack at the same snowpit site at one point in time. The snowpit analysis used in this study focused on the physical properties of the snowpack, including depth, density, anisotropy, and thermal conductivity. In this study, we analyze the MOSAiC snowpit dataset (Macfarlane et al., 2021a), of which three key instruments were the focus of this study. The three instruments included in this study were (i) $\mu$CT, (ii) SMP, and (iii) a density cutter. Details of these instruments are given below, and an example of the snowpit site set-up can be seen in Fig. 1.

### 2.2 $\mu$CT samples

The dataset evaluated for this paper includes 138 $\mu$CT samples (approximately 10 cm high and 6.6 to 7.8 cm in diameter) collected during 69 individual visits to the sea ice, known as "Events". More than one $\mu$CT sample was often collected during an event to sample the complete snow profile. The EventID (a unique labeling system representing one trip to the ice) can identify co-located $\mu$CT samples. A three-dimensional reconstruction of two $\mu$CT samples is given in Fig. 1b. A schematic of the location of three $\mu$CT samples taken from the event with EventID PS122-3_35-56 can be seen in Fig. 1c. The snow samples were extracted using an electric cylindrical drill, carefully placed in a sample holder, and transported back to the laboratory on *Polarstern* (Knust, 2017). By installing a desktop cone-beam microCT 90 ($\mu$CT) in a laboratory onboard, we could measure the microstructure of the snow semi-in situ. The laboratory was cooled to $-15\,°C$, and the $\mu$CT had a custom ventilation system meaning the sample remained at $-12\,°C$ during the scanning process.

Once the snow samples were scanned, the data were analyzed by dividing each snow sample into sub-samples of volume 5.83 cm$^3$ (18 mm × 18 mm × 18 mm) to calculate the density and the geometrical anisotropy defined by

$$A_{\mathrm{g}} = \frac{2\xi_z}{\xi_x + \xi_y}, \tag{1}$$

in terms of the correlation lengths $\xi$ in different coordinate directions $x, y, z$. The correlation lengths were obtained by fitting the decay of the two-point correlation function in different directions to an exponential (Löwe et al., 2013). Subsequently, the effective thermal conductivity was computed through FEM.

Microstructure-based FEM is a standard method for computing the effective thermal conductivity tensor of two-phase materials, which governs macroscopic heat flow on length scales that are large compared to the microstructural scales of the ice matrix. Here we have used the finite element code (Garboczi, 1998), which solves the variational formulation of the conduction problem with periodic boundary conditions. The numerical simulations carried out here precisely follow the procedures described by Löwe et al. (2013) and Gouttevin et al. (2018).

We computed the effective thermal conductivity tensor **k** (W K$^{-1}$ m$^{-1}$) from the 138 3-D $\mu$CT sample images collected throughout winter during the MOSAiC expedition, as outlined above, following Calonne et al. (2011); Löwe et al. (2013). For the thermal conductivity of ice ($k_{\mathrm{ice}}$) and air ($k_{\mathrm{air}}$), we used their values at $T = -20\,°C$, namely $k_{\mathrm{ice}} = 2.34$ W K$^{-1}$ m$^{-1}$ (Slack, 1980) and $k_{\mathrm{air}} = 0.024$ W K$^{-1}$ m$^{-1}$, and followed Calonne et al. (2019), who referenced Paterson (2000) for the ice conductivity values and Yen (1981) for the air conductivity values. We assume transverse isotropy in the (horizontal) $xy$ plane, which is reasonable when temper-

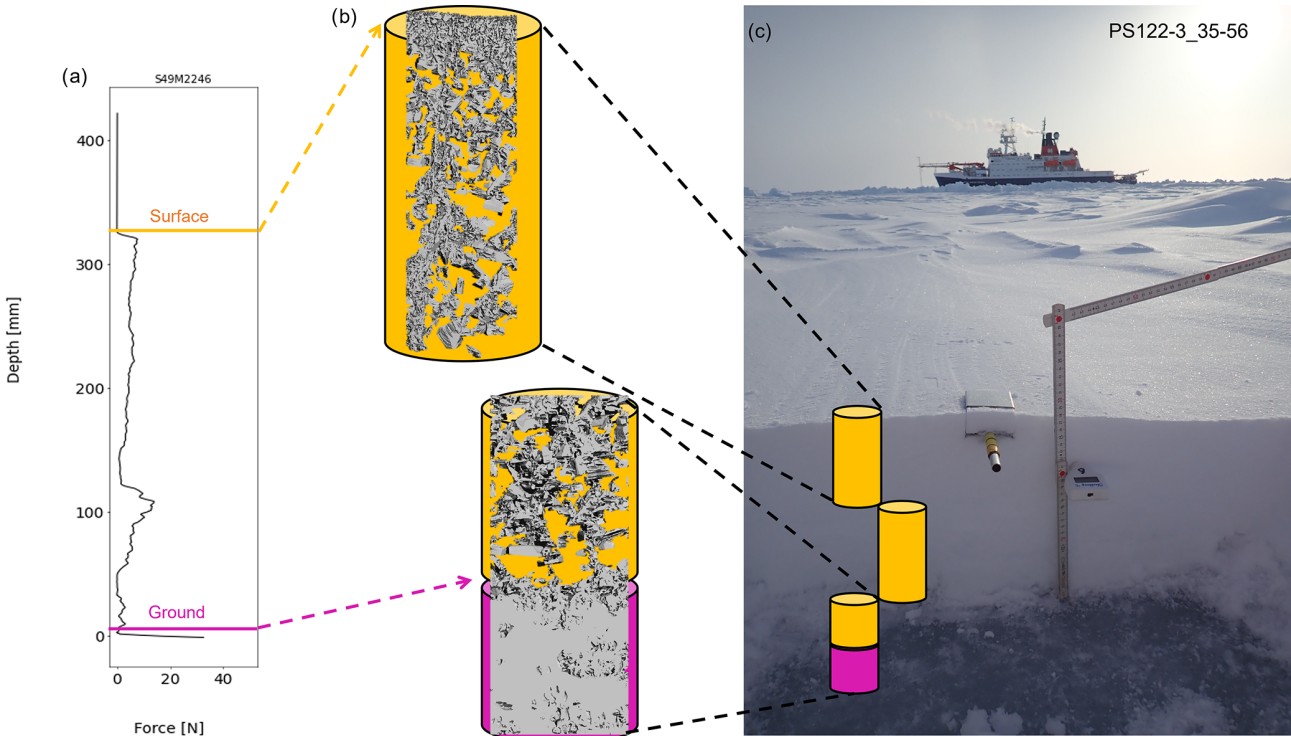

**Figure 1.** Overview schematic of the snowpit set-up. **(a)** A snow micro penetrometer (SMP) force signal showing stratigraphy of the snowpack during Event PS122-3_35-56. **(b)** Three-dimensional reconstructions of two $\mu$CT samples showing a typical surface (top) and snow–sea-ice interface (bottom) sample. The yellow background indicates a region classified as snow (density $< 550\,\mathrm{kg\,m^{-3}}$), and the pink background indicates a sample including ice (with densities $> 550\,\mathrm{kg^{-3}}$). **(c)** The overview photo of the snowpit during Event PS122-3_35-56.

ature gradients are aligned with the (vertical) $z$ direction. In this coordinate system, the tensor is diagonal, and we refer to $k_{\mathrm{eff},z}$ as the vertical component and to $k_{\mathrm{eff},xy}$ (the average of $k_{\mathrm{eff},x}$ and $k_{\mathrm{eff},y}$) as the horizontal component of the effective thermal conductivity tensor.

The so-obtained effective thermal conductivity ($k_{\mathrm{eff},z}$) characterizes the steady-state, conductive heat flow through a unit area of a homogeneous material induced by a unit temperature gradient in a direction perpendicular to that unit area ($\mathrm{W\,K^{-1}\,m^{-1}}$). In the following we mostly focus on vertical temperature gradients and denote $k_{\mathrm{eff},z}$ by $k_{\mathrm{eff}}$ throughout this study. The relationship is shown in Eq. (2), where $h$ is the sample thickness (m), $\Delta T$ represents the temperature difference (K), and $q$ represents the volume averaged heat flux ($\mathrm{W\,m^{-2}}$).

$$k_{\mathrm{eff}} = q\frac{h}{\Delta T} \tag{2}$$

The thermal conductivity tensor was also used to calculate the thermal anisotropy $A_k$ of the samples (Eq. 3) (Calonne et al., 2011; Riche and Schneebeli, 2013):

$$A_k = \frac{k_{\mathrm{eff},z}}{k_{\mathrm{eff},xy}}. \tag{3}$$

The thermal anisotropy $A_k$ is largely correlated with the geometrical anisotropy $A_g$ (see Appendix and Löwe et al., 2013).

The thermal conductivity of the $\mu$CT sub-samples, calculated from FEM ($k_{\mathrm{eff}}^{\mathrm{FEM}}$), were then analyzed for the parameterizations in view of density and the thermal and geometrical anisotropy of the sub-samples, to identify sources of variability.

### 2.2.1 Parameterizations of thermal conductivity

To distinguish different parameterizations for the effective conductivity, we use the notation $k_{\mathrm{eff}}^{\mathrm{P}}$, where P represents a particular formulation. For details on the difference between each parameterization, please refer to Table 1, adapted from Calonne et al. (2019).

The $k_{\mathrm{eff}}^{\mathrm{P}}$ parameterizations tested in this study used (a) density, (b) density and temperature, and (c) density and anisotropy. An overview is given in Table 1.

A temperature of $-20\,^\circ\mathrm{C}$ was used in the density and temperature parameterizations, as this was representative of the temperature conditions throughout the winter during the MOSAiC expedition (more details are given in Sect. 2.5). Our simulations use $k_{\mathrm{ice}}$ at $-20\,^\circ\mathrm{C} = 2.34\,\mathrm{W\,K^{-1}\,m^{-1}}$, and we chose to analyze the Calonne et al. (2019) parameter at $-20\,^\circ\mathrm{C}$.

By comparing these parameterizations to the values of $k_{\mathrm{eff}}^{\mathrm{FEM}}$, we could identify which parameters are optimal for

**Table 1.** An overview of the thermal conductivity parameterizations. An overview of the thermal conductivity parameterizations used throughout this paper from Yen (1981), Sturm et al. (2002a), Calonne et al. (2019), and Löwe et al. (2013).

| P | Formula | Density | Temperature | Anisotropy |
|---|---------|---------|-------------|------------|
| Yen | $k_{\text{eff}}^{\text{Yen}} = 2.22362\left(\frac{\rho}{1000}\right)^{1.885}$ | 80–600 | Undefined | No |
| Stm | $k_{\text{eff}}^{\text{Stm}} = 0.023 + 0.234\frac{\rho}{1000}$ | for $\rho < 156$ | Average $-15\,^{\circ}\text{C}$ | No |
| | $= 0.138 - 1.01\frac{\rho}{1000} + 3.233\left(\frac{\rho}{1000}\right)^2$ | for $156 < \rho < 600$ | | |
| Cal20 | See Calonne et al. (2019) | 102–888 | $-20\,^{\circ}\text{C}$ | No |
| Löwe | See Löwe et al. (2013) | Approx. 91.6–460 | $-20\,^{\circ}\text{C}$ | Yes |

measuring $k_{\text{eff}}^{\text{P}}$ for snow on Arctic sea ice. After conducting this analysis, we calculated the second-order polynomial fit for this dataset to obtain a density parameterization specific for snow on sea ice, as seen in Eq. (4), where $\rho$ represents the density of the sub-samples, $a = 2.62 \times 10^{-6}$, $b = 1.54 \times 10^{-33}$, and $c = 3.04 \times 10^{-2}$.

$$k_{\text{eff}}^{\text{Mac(I)}} = a\rho^2 + b\rho + c \tag{4}$$

When additionally allowing for anisotropy in the parameterization, it is straightforward to generalize Löwe et al. (2013) to obtain an accurate parameterization as a function of density and $A_{\text{g}}$ in the entire density range. This parameterization is denoted by

$$k_{\text{eff}}^{\text{Mac(II)}} = k_0 + k_{\text{ice}}\left(\frac{X^{\beta}}{\Omega(1 - X) + X^{(\beta - 1)}},\right) \tag{5}$$

with $X = k_z^{(\text{L})}/k_{\text{ice}}$; free parameters $k_0$, $\beta$, and $\Omega$; and known function $k^{(\text{L})}$. The motivation and details for Eq. (5) are given in the Appendix.

## 2.3 SMP profiles

The snow micro penetrometer (SMP) instrument measures the penetration force resistance of a snow profile at 0.3 mm vertical resolution. Five SMP force profiles were obtained within one snowpit, approximately 0.25 m apart. Additional measurements were often taken on both sides of the snowpit to capture the spatial heterogeneity of the snow in the surrounding area. These additional SMP measurements were taken at intervals of 1 m, which reduced operator bias when selecting an area to measure. More details of the measurement protocol can be found alongside the published dataset and data paper (Macfarlane et al., 2021b).

Additionally, further details on the dataset can be found alongside the published SMP data (Macfarlane et al., 2021b) within the snowpit bundle (Macfarlane et al., 2021a). A total of 3266 SMP profiles are used in this study. The SMP penetration force profile can be used to obtain density and (in combination with parameterizations listed in the previous Section) estimates of the thermal conductivity. To obtain density from the force profile, we used the density parameterization from King et al. (2020). The seasonal comparison

of the density obtained by these instruments can be seen in Fig. 7. This parameterization was chosen because the dataset was also collected on sea ice in the high Arctic, meaning similar snow grain types were measured Kaltenborn et al. (2023). When comparing the snow density using (a) a density cutter to (b) density derived from the SMP and the King et al. (2020) parameterization, we experienced difficulties using the field data due to the high spatial heterogeneity at the meter scale. Comparing the field measurements taken just a few centimeters apart showed different stratigraphy profiles. This is the primary challenge when measuring snow in the snow–sea-ice landscape. We try to answer the following questions. How do we measure a representative sample size? How do we understand what variability is due to the uncertainty of our measurement methods, and what is the result of the spatial heterogeneity? To derive an uncertainty, further laboratory work (by using similar methods to Riche and Schneebeli, 2013) is needed to understand the uncertainties of the SMP-density-derived method.

### 2.3.1 The effective thermal conductivity's harmonic mean

As stated before, we assume that the thermal gradient in a snowpack is vertical. For a layered material, such as snow, the average thermal conductivity for the entire snowpack must take the layering into account. This average thermal conductivity can be calculated in analogy to Ohm's law by conduction resistances in series. The harmonic mean of a snow profile's thermal resistance ($k_{\text{eff}}^{\text{P}}$) is calculated using Eq. (6), where $n$ is the number of sub-samples in a profile, and $k_i$ is the effective thermal conductivity of individual sub-samples (simplified as all sub-samples have the same dimension).

$$\overline{k_{\text{eff}}^{\text{P}}} = \left(\frac{\sum_{i=1}^{n} k_i^{-1}}{n}\right)^{-1} \tag{6}$$

After testing the listed parameterizations in Table 1, we used the parameterization with the highest $r^2$ in relation to this dataset to upscale the single snowpits. The harmonically

averaged $\overline{k_{\text{eff}}^{\text{P}}}$ of all the SMP profiles in winter were then grouped depending on the snowpit site's underlying ice type (e.g., FYI areas, SYI areas, or refrozen leads), topographic features (e.g., ridges), and month to understand spatial heterogeneity better.

### 2.3.2   Average effective thermal resistance

The SMP measurements of thermal conductivity and snow depth were used to investigate the snow's thermal resistance ($R$) on the ice floe using the $k_{\text{eff}}^{\text{Mac(I)}}$ parameterization. We conducted tests to see whether $R$ is directly proportional to HS (snow depth) or if $\overline{k_{\text{eff}}}$ also has an influence on the snow profile's thermal resistance. The snowpack's $R$ value is the temperature difference, at steady state, between the ice–snow interface and ice–atmosphere interface, given a unit heat flow rate through a unit area ($\text{m}^2\,\text{K}\,\text{W}^{-1}$). By combining this definition and Eq. (2), the snowpack's $R$ can be found by dividing the snow depth (HS) by the profile's $\overline{k_{\text{eff}}^{\text{P}}}$, as seen in Eq. (7). Thermal resistance is a useful parameter for modeling heat transfer in the sea ice system as it relates to snow thermal conductivity and depth. If snow is considered as an interface between the atmosphere and the sea ice in models, it is beneficial to use the reciprocal of the thermal conductivity multiplied by the layer thickness rather than a conductivity. This is explained nicely in Bigdeli et al. (2020) using an analogy of a simple electrical circuit. An extract from Bigdeli et al. (2020) is given below.

> Consider electrical resistors, which, when placed in series, carry the same current. Similarly, our ice and snow layers convey the same vertical heat flux sequentially. The total resistance of the electrical resistors in series is simply the sum of their individual resistances. Analogously, the snow and ice resistances in our system are additive but their (reciprocal) conductivities are not. The resistance of snow per meter ($3.22\,\text{W}^{-1}\,\text{m}\,\text{K}$) is approximately 7 times larger than that of ice per meter ($0.46\,\text{W}^{-1}\,\text{m}\,\text{K}$). Considering a case where 10 cm of snow is lost through surface melt as an example, it is now easy to see that 70 cm of ice would need to form via basal freezing in order to retain the same total insulating effect, highlighting the efficiency of snow as a thermal buffer.

The resistance is, therefore, beneficial to Arctic climate simulations without explicitly resolving the snow cover. We tested the dependence of thermal resistance on the underlying ice type. We initially assumed a thermal conductivity and snow height dependence on the underlying ice type, as mentioned in Nicolaus et al. (2009). To test this, the measurements were grouped as mentioned in Sect. 2.3.1.

$$R = \frac{\Delta T}{q} = \frac{\text{HS}}{\overline{k_{\text{eff}}^{\text{P}}}} \tag{7}$$

### 2.4   Density profiles

We investigated temporal changes in thermal conductivity using all density measurements available in the winter period. The instruments that are used to measure density include a density cutter ($\rho^{\text{Cutter}}$), a snow water equivalent (SWE) tube ($\rho^{\text{SWE}}$; measuring snow water equivalent), and the SMP measurements ($\rho^{\text{SMP}}$), using the parameterization from King et al. (2020) as indicated in Sect. 2.3.

### 2.5   Atmospheric data

Using three independent instruments, we investigated the influence of atmospheric conditions on the seasonal evolution of snow density and thermal conductivity. We analyzed shortwave radiation data (Riihimaki, 2021) from up and down radiometer systems, temperature, and wind data measured at 2 m (Cox et al., 2021) from a meteorological flux tower. These instruments were deployed at Met City (a station approximately 200 m away from the snowpit measurements). This additional atmospheric data helped us understand and explain the conditions that might influence the density and thermal conductivity of the snow cover. We also used the atmospheric conditions to confirm using the Calonne et al. (2019) parameterization at $-20\,°\text{C}$.

## 3   Results

### 3.1   Microstructure-based FEM

Individual vertical snow profiles showed high vertical variability in (a) the density profiles and (b) $A_k$ and, as a result of this, high variability in $k_{\text{eff}}^{\text{FEM}}$. Icy layers within the snow profile, crusts on the surface, and a "remnant surface scattering layer" at the snow–ice interface (a granular layer at the top of the melting summer sea ice (Macfarlane et al., 2023a)) were of high density and low $A_k$, in contrast to the low-density precipitated snow and high values of $A_k$ for depth hoar. The vertical profiles of $k_{\text{eff}}^{\text{FEM}}$ in Fig. 2 highlight the large variability amongst samples, showing that snow stratigraphy highly influences thermal conductivity.

The commonly occurring layers of depth hoar and rounded, wind-blown snow are of similar densities of approximately $300\,\text{kg}\,\text{m}^{-3}$. Due to these two grain types being dominant on Arctic sea ice, we see a large proportion of our sub-sample's densities in the range of 200 to $400\,\text{kg}\,\text{m}^{-3}$, seen in the high point concentration in this density range in Fig. 3. The color in this figure shows the range of $A_k$ values and the influence of $A_k$ on $k_{\text{eff}}^{\text{FEM}}$. $A_k$ values ranged between 0.25 and 2, indicated in the legend in Fig. 3. The extreme anisotropy values in the lower range show icy layers, and the high values are depth hoar samples.

The density distribution of the $k_{\text{eff}}^{\text{FEM}}$ values are shown in Fig. 4, after a $550\,\text{kg}\,\text{m}^{-3}$ density cut-off is applied. This threshold was chosen as we found some wind-packed, depth

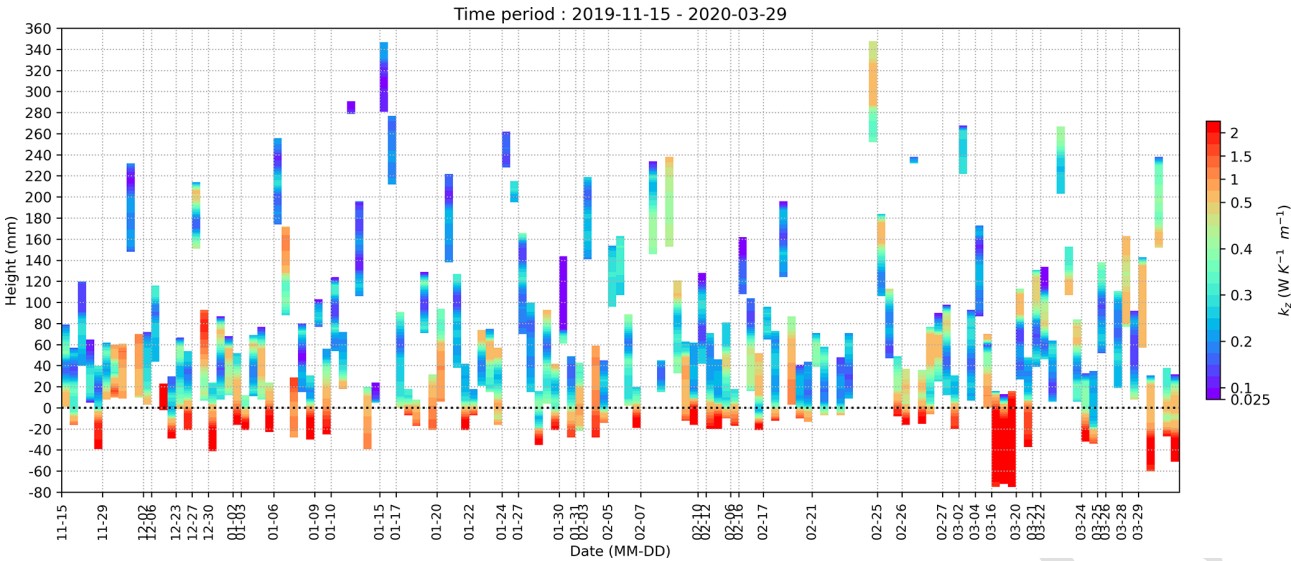

**Figure 2.** Samples of effective thermal conductivity plotted against time. Snow samples were collected during the winter to be measured using micro-computed tomography. We simulated effective thermal conductivity across these samples using microstructure-based FEM. Here, we see each sample plotted at the height taken in the snowpack against the collection date. Negative heights correspond to sea ice samples beneath the snowpack, which are excluded from any snow thermal conductivity calculations. This figure highlights the vertical variability within the samples.

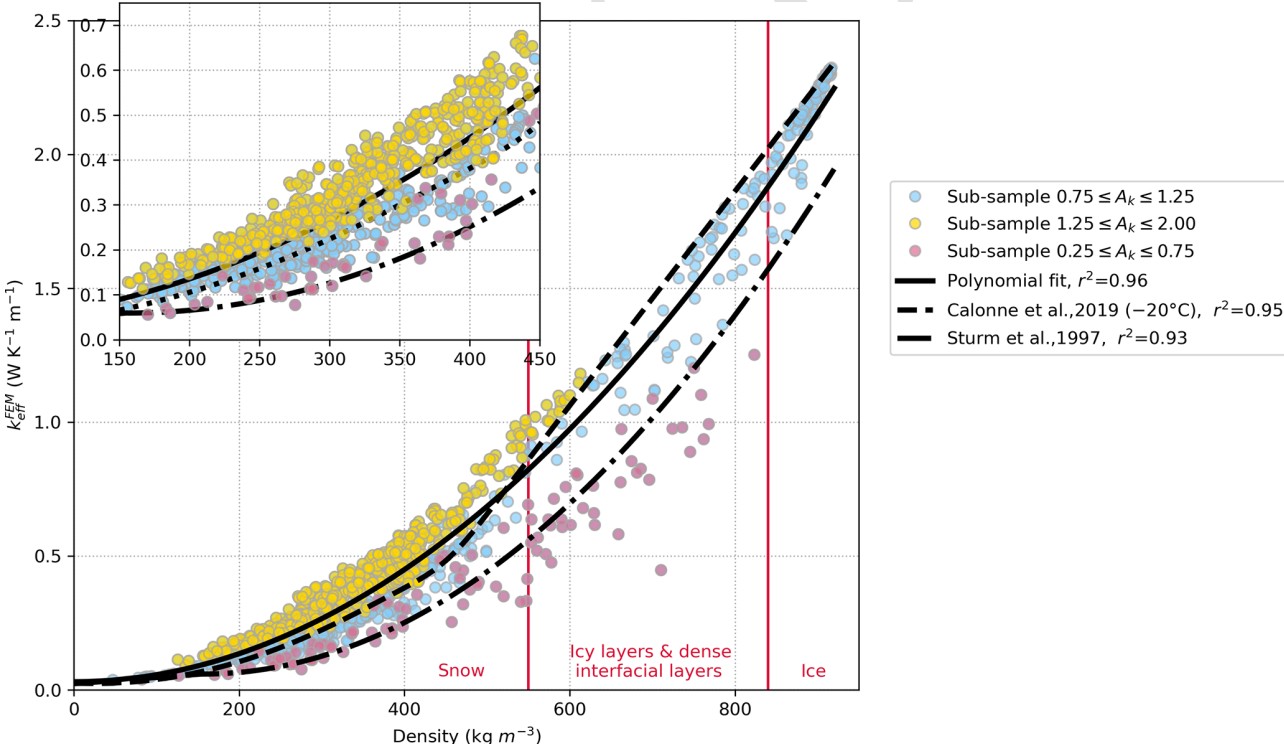

**Figure 3.** The sub-sample density plotted against effective thermal conductivity. Microstructure-based FEM of the effective thermal conductivity for the sub-samples is compared to the sub-sample density. A polynomial fit of the data is shown in the solid line. This relationship between effective thermal conductivity and density has been tested in previous studies. This figure includes two current parameterizations (Calonne et al., 2019; Sturm et al., 1997). Anisotropy values are indicated in different colors, with details given in the legend, and the figure shows how anisotropy influences the effective thermal conductivity of the sub-samples. The vertical red lines represent the cut-off between snow, icy layers in the snowpack, and sea ice.

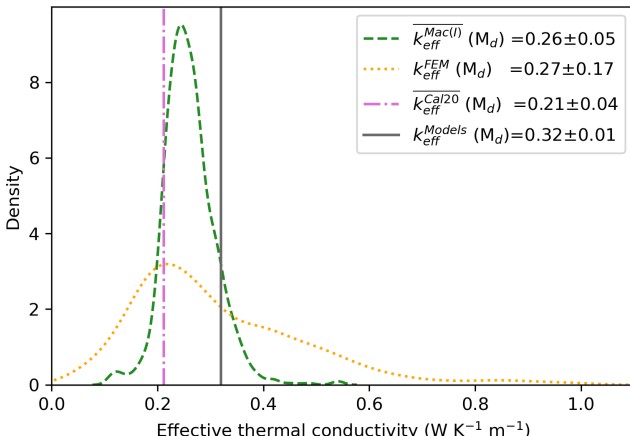

**Figure 4.** Density plot of effective thermal conductivity. The density distribution of $k_{\text{eff}}^{\text{FEM}}$ for all $\mu$CT sub-samples with densities below $500\,\text{kg}\,\text{m}^{-3}$ and the harmonic mean of the SMP profiles ($\overline{k_{\text{eff}}^{\text{Mac(I)}}}$) from January to March 2020. The legend indicates the median values with the symbol $M_d$. The error given in the legend represents 1 standard deviation.

hoar snow layers to have a high density, with values ranging up to $550\,\text{kg}\,\text{m}^{-3}$. In addition, we wanted to exclude ice samples (Britannica, 2014) and the hard interfacial layers found on second-year ice (mentioned above as a possible "remnant surface scattering layer"). The average $k_{\text{eff}}^{\text{FEM}}$ value is $0.27 \pm 0.17\,\text{W}\,\text{K}^{-1}\,\text{m}^{-1}$. The errors given throughout this paper are 1 standard deviation ($\pm 1\sigma$).

### 3.2 Parameterizations of thermal conductivity

The high sample variability allowed our dataset to cover density values of approximately 50 to $950\,\text{kg}\,\text{m}^{-3}$ and anisotropy values between 0.25 and 2. This allowed us to test each $k_{\text{eff}}^{\text{P}}$ parameterization presented in this paper. When comparing $k_{\text{eff}}^{\text{P}}$ to $k_{\text{eff}}^{\text{FEM}}$ for all sub-samples, Fig. 5 shows the relationship for current parameterizations for the full range of possible snow densities. The $r^2$ values for each parameterization analyzing the entire dataset can be found in Fig. 5. However, some parameterizations result in a low $r^2$ value due to the adjustable coefficients in the original work being optimized only in specific density ranges. These are outlined in Table 2. For this reason, we conducted mean absolute error (MAE) tests on the dataset with different thresholds (density thresholds set to below and above $550\,\text{kg}\,\text{m}^{-3}$). The results can be seen in Table 2.

Without including anisotropy in the parameterization, $k_{\text{eff}}^{\text{Mac(I)}}$ is the best representation of $k_{\text{eff}}$ for the entire dataset, as it has the highest $r^2$ value compared to the microstructure-based FEM dataset. We use this parameterization and introduce the SMP to upscale our measurements of $k_{\text{eff}}$, of which we do not have corresponding $A_k$ or $A_g$ measurements for this study. Anisotropy is critical for reducing uncertainty in

**Table 2.** Statistical tests for each parameterization. Mean absolute error (MAE) analysis conducted at different density $\rho$ ($\text{kg}\,\text{m}^{-3}$) thresholds for each parameterization ($P$) presented in this study alongside the $r$-squared value of the entire range of density values for this dataset (approximately 50–$900\,\text{kg}\,\text{m}^{-3}$).

| Parameterization ($P$) | $r^2$ (entire dataset) | MAE ($50 < \rho < 550$) | MAE ($\rho > 550$) |
|---|---|---|---|
| Yen | 0.89 | 0.07 | 0.31 |
| Stm | 0.82 | 0.15 | 0.32 |
| Mac(I) | 0.97 | 0.05 | 0.15 |
| Cal20 | 0.96 | 0.07 | 0.15 |
| Löwe | 0.27 | 0.03 | 4.40 |
| Mac(II) | 0.99 | 0.03 | 0.06 |

thermal conductivity; this is mentioned again in the discussion, and future work is suggested.

### 3.3 Spatial heterogeneity and temporal changes

For the rest of the study, we use SMP profiles and the effective thermal conductivity's harmonic mean, $\overline{k_{\text{eff}}^{\text{Mac(I)}}}$, using the density of the SMP profiles ($\rho^{\text{SMP}}$), calculated using the (King et al., 2020) parameterization of density to investigate spatial heterogeneity and temporal changes of the snow cover on Arctic sea ice.

To understand the heterogeneity of the snow depth (HS), we categorized the snowpits in situ into ice type and ridged areas. Figure 6 shows the snow heights, snow density (measured using the SMP and the King et al., 2020 parameterization), thermal conductivity, and thermal resistance for each ice type and for ridge areas. This can be seen in the grey box plots in the background of Fig. 6. Table 3 shows that more snow is found on ridges with HS $= 335\,\text{mm}$ and less on leads (as this ice type is when thin ice has formed and snow has just started to accumulate), with $84\,\text{mm}$ on average. A breakdown of this dataset to investigate the average of each parameter for individual months can be seen in the colored bar charts in Fig. 6. The snow depth is highly variable on all ice types, with standard deviations of between $109\,\text{mm}$ on FYI and $278\,\text{mm}$ on ridges. The range of snow depth on ridges (0 to $> 1000\,\text{mm}$) shows consistently high spatial heterogeneity throughout the winter season; therefore, temporal changes are less discernible than in FYI and SYI areas.

The snow density ($\rho^{\text{SMP}}$) median is slightly higher on refrozen leads, FYI, and ridges compared to snow densities on SYI (values are given in Table 3). Snow density has a similar monthly trend on all ice types (shown in the colored box plots in Fig. 6), increasing until February/March CEI and then decreasing in April. Looking at the median density values for this season in Fig. 7 shows this feature in multiple datasets, not just the SMP. Figure 7 shows a density increase from November to March ($\rho^{\text{SMP}}$ increases by $43\,\text{kg}\,\text{m}^{-3}$, $\rho^{\text{Cutter}}$ increases by $78\,\text{kg}\,\text{m}^{-3}$, and $\rho^{\text{SWE}}$ in-

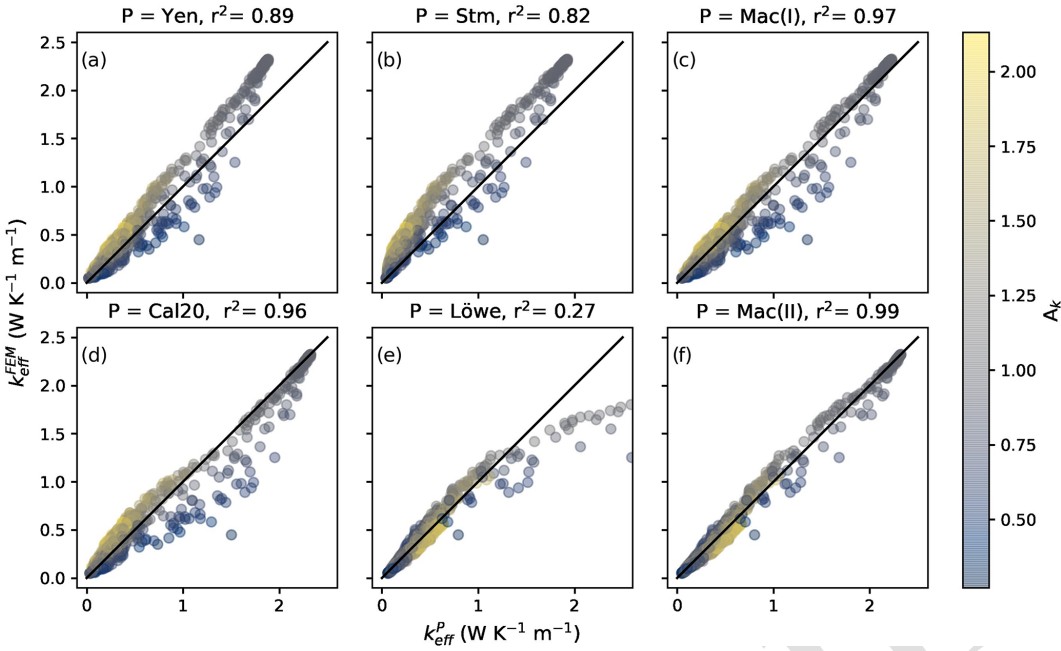

**Figure 5.** Different parameterizations of effective thermal conductivity plotted against effective thermal conductivity measured using microstructure-based FEM. Effective thermal conductivity was measured for each sub-sample using (i) the microstructure-based finite element method (FEM) and (ii) different parameterizations using density, anisotropy, and temperature. Panel **(a)** shows the performance of density parameterizations in Yen (1981), panel **(b)** the density parameterization by Sturm et al. (1997), and panel **(c)** the polynomial fit of this density dataset $k_{\mathrm{eff}}^{\mathrm{Mac(I)}}$. Panel **(d)** uses the temperature approximation by Calonne et al. (2019) at $-20\,°C$. Panel **(e)** shows the performance of the anisotropy and density parameterization by Löwe et al. (2013), plotted against the FEM-measured effective thermal conductivity. Finally, panel **(f)** shows the optimization of the anisotropy and density parameterization, presented in this study as $P = \mathrm{Mac(II)}$ in Eq. (5).

**Table 3.** Snow depth, density, thermal conductivity, and resistance for each ice type. The median $(\pm1\sigma)$ of snow depth (HS), density $(\rho^{\mathrm{SMP}})$, harmonically averaged effective thermal conductivity using the Mac(I) parameterization $(\overline{k_{\mathrm{eff}}^{\mathrm{Mac(I)}}})$, and thermal resistance $(R)$ for each ice type.

| Ice type | HS (mm) | $\rho^{\mathrm{SMP}}$ (kg m$^{-3}$) | $\overline{k_{\mathrm{eff}}^{\mathrm{Mac(I)}}}$ (W K$^{-1}$ m$^{-1}$) | $R$ (m$^2$ K W$^{-1}$) |
|---|---|---|---|---|
| Refrozen leads | $84 \pm 124$ | $301 \pm 41$ | $0.25 \pm 0.06$ | $350 \pm 469$ |
| FYI | $129 \pm 109$ | $294 \pm 32$ | $0.24 \pm 0.05$ | $515 \pm 404$ |
| SYI | $144 \pm 113$ | $277 \pm 26$ | $0.22 \pm 0.04$ | $660 \pm 475$ |
| Ridges | $335 \pm 278$ | $304 \pm 30$ | $0.26 \pm 0.05$ | $1411 \pm 910$ |

creases by $96\,\mathrm{kg\,m^{-3}}$) and a decrease after that (average $\rho$ decrease in April is $24.3\,\mathrm{kg\,m^{-3}}$). The SMP penetration resistance was normalized for the snow depth (Fig. 8) to better see changes throughout the season. Figure 8 shows a surface snow density increase in March followed by a reduction in April at the surface and lower depths of the snow cover. This is further discussed in Sect. 4.3.

$k_{\mathrm{eff}}^{\mathrm{Mac(I)}}$ has a standard deviation of between 0.04 and $0.06\,\mathrm{W\,K^{-1}\,m^{-1}}$ for all ice types; the difference between the median $\overline{k_{\mathrm{eff}}^{\mathrm{Mac(I)}}}$ of these ice types is $0.04\,\mathrm{W\,K^{-1}\,m^{-1}}$. These data can be found in Table 3. We see that $\overline{k_{\mathrm{eff}}^{\mathrm{Mac(I)}}}$ has a slight increase until March and a decrease thereafter. We excluded

any measurements conducted in May 2020, as the number of measurements was insufficient to draw any conclusions on the temporal trend.

Due to $\overline{k_{\mathrm{eff}}}$ having no significant variability on different ice types (3), we can state that $R$ is directly proportional to HS. The average $R$ for the winter is lowest on refrozen leads and FYI areas, increasing slightly on SYI and highest on ridged areas. $R$ remained constant through the season on FYI and SYI. Refrozen leads and ridges had high variability between months.

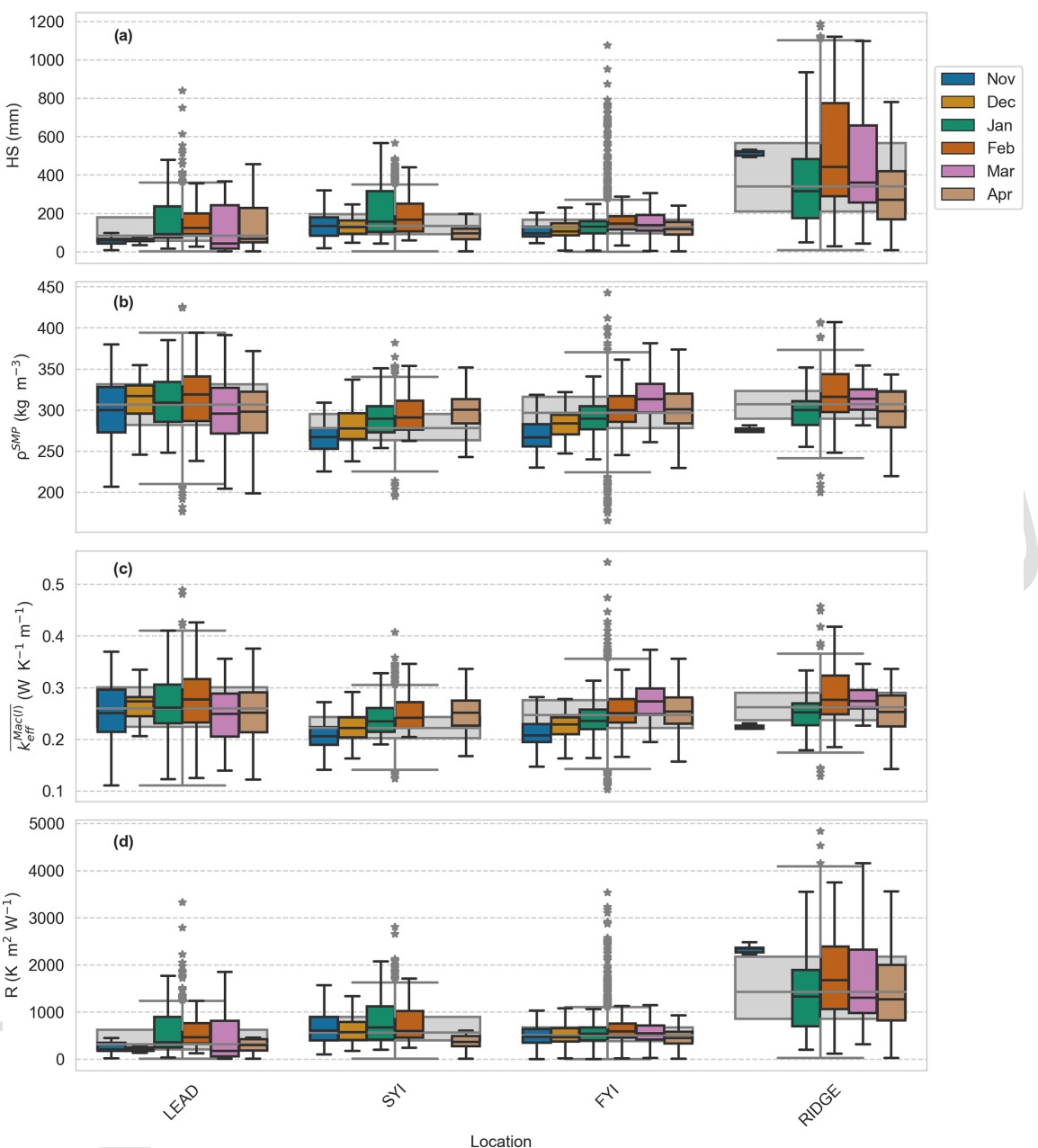

**Figure 6.** A box plot showing spatial and temporal trends for measured snow parameters. Snow depth (HS), density ($\rho^{SMP}$), harmonically averaged thermal conductivity ($\overline{k_{eff}^{Mac(I)}}$), and resistance ($R$) were all measured using a snow micro-penetrometer and plotted against underlying ice type and month. A snow micro penetrometer was used to measure vertical profiles of penetration resistance. These profiles can be used to extract snow depth; density (using King et al., 2020); harmonically averaged effective thermal conductivity using the Mac(I) parameterization, given in Eq. (4), with the King et al. (2020) derived density as an input; and, finally, the resistance of the snowpack ($R$). These profiles are grouped by underlying ice type, topographic feature (seen in the grey bar charts in the background of the figure, with grey stars indicating the outliers), and month (seen in the colored bar charts, of which the outliers are not shown).

## 4   Discussion

Before advancing our understanding of the snow's thermal conductivity heterogeneity and temporal trends, we must assess the performance of existing parameterizations on samples of snow measured on sea ice in the high Arctic. The

$\mu$CT simulations allowed us to assess the current parameterizations for the complete ranges of density and anisotropy values. Following this, we introduced two new parameterizations with and without anisotropy, specifically adapted for the use of snow on sea ice. The $\mu$CT is highly time-

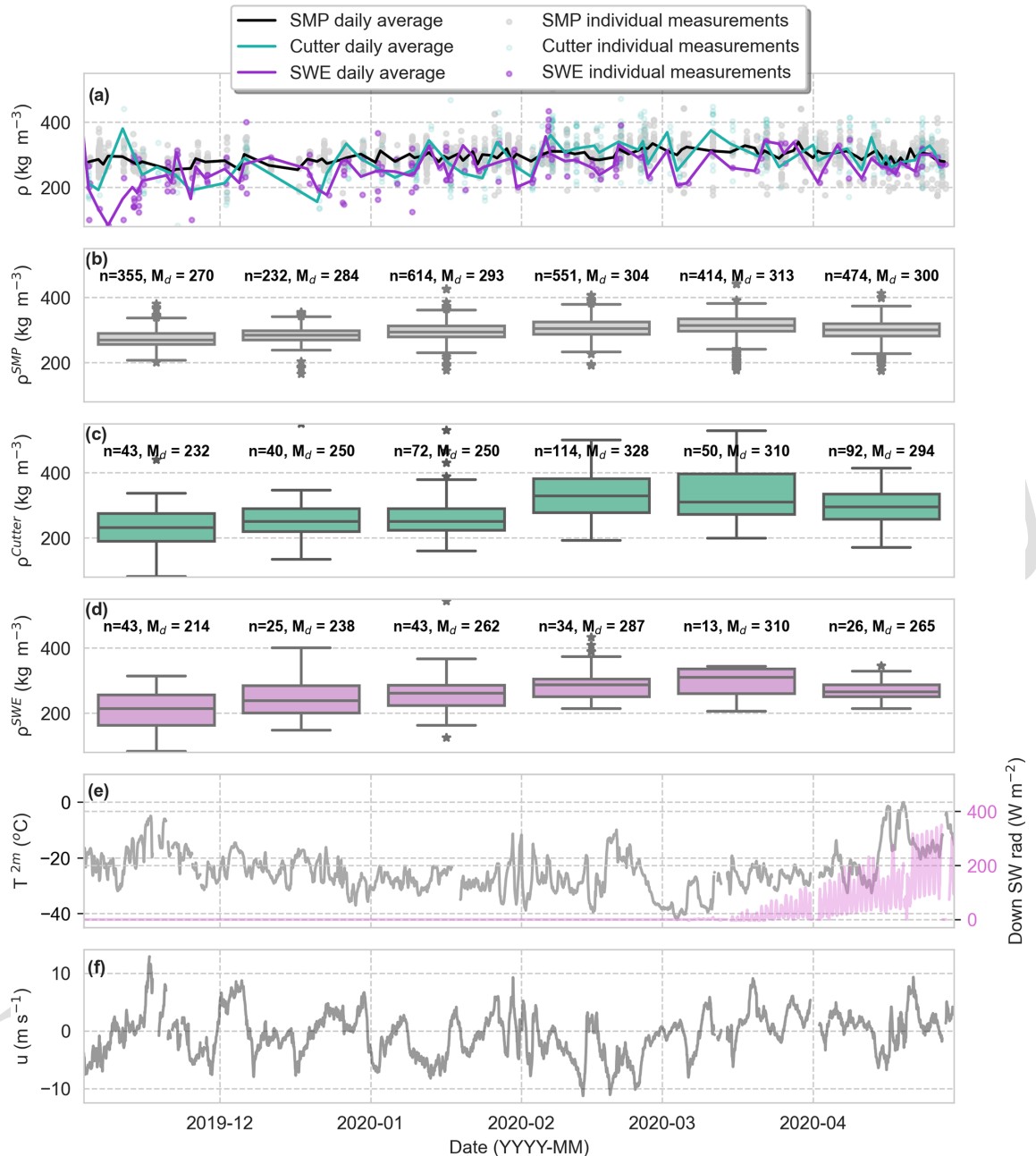

**Figure 7. (a)** Time series of density using three independent instruments. The lines show the daily average, and the points show the individual measurements. **(b)** A box plot grouping the snow micro penetrometer density measurements by month. **(c)** A box plot grouping the density cutter density measurements by month. **(d)** A box plot grouping the snow water equivalent density measurements by month. All box plots show the temporal change in the medians ($M_d$) and the number of data points in each box plot ($n$). **(e)** The local air temperature at 2 m above the snow surface ($T^{2m}$) and downward shortwave (SW) radiation. **(f)** Time series of wind speeds ($u$). Density measurements from different instruments within the snowpit are compared in the upper plot against time.

demanding, so to investigate the spatial variability of the snow cover we introduced the SMP to have more measurements. The SMP does not currently have anisotropy measurements in parallel; therefore, the density parameterization ($k_{\mathrm{eff}}^{\mathrm{Mac(I)}}$, given in Eq. 4) was used for this upscaling, as it had the highest $r^2$ value for this dataset when compared to

the $k_{\mathrm{eff}}^{\mathrm{FEM}}$ values. Future SMP measurements, in combination with methods seen in Kaltenborn et al. (2023), hold the possibility of deciphering the anisotropy of the snow grains in the field using the SMP. If a grain type is classified through SMP profiles (using the methods in Kaltenborn et al., 2023), then we could approximate the anisotropy of these different grain

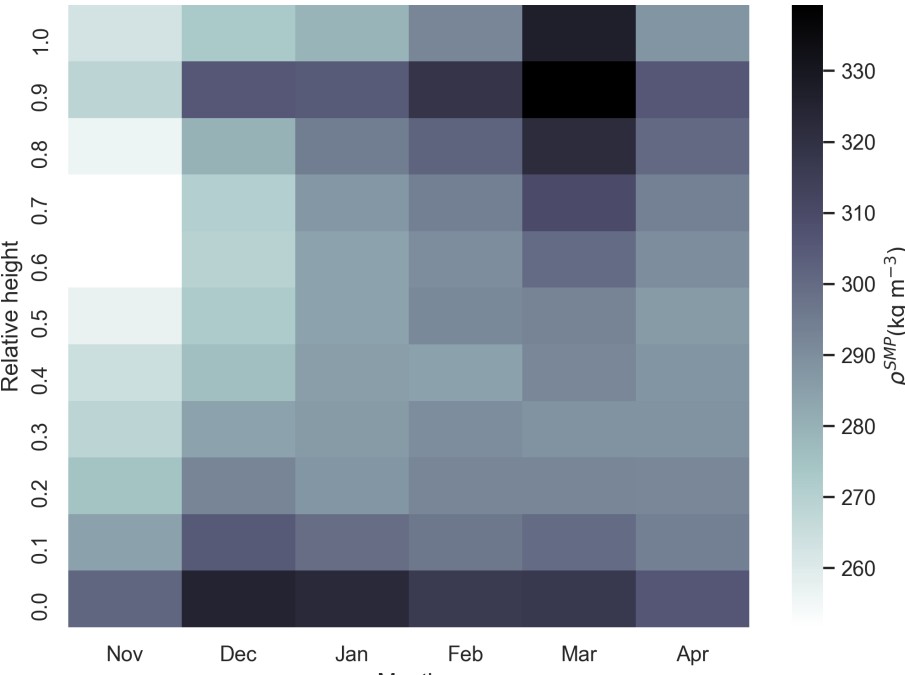

**Figure 8.** A heatmap of winter snow micro-penetrometer profiles on level FYI and SYI. All snow micro penetrometer profiles are concatenated, and their depths are normalized. The normalized SMP density signals averaged for all profiles within 1 month are displayed throughout the winter to show seasonal changes in the snow cover. A relative height of 0 represents the snow–ice interface, and 1 represents the snow–air interface. The denser snow surface in March shows higher thermal conductivity values, possibly due to storm events with high wind speeds.

classes and improve thermal conductivity measurements using the SMP. This is explained in greater detail below.

## 4.1 Assessing existing parameterizations

A large range of the sub-sample density and anisotropy values allowed us to create parameterizations of thermal conductivity (and test existing parameterizations) for all ranges of density (from 50 to 900 $kg\,m^{-3}$) and anisotropy (from 0.25 to 2). The relationship between density and $k_{eff}^{FEM}$ in Fig. 3 was compared to the parameterizations from Calonne et al. (2019) at $-20\,°C$ and Sturm et al. (1997). Through this comparison, we can see that the anisotropy heavily influences the $k_{eff}^{FEM}$ values. For example, a snow sub-sample with a density of $400\,kg\,m^{-3}$ can have a thermal conductivity value ranging from 0.2 to 0.6 $W\,K^{-1}\,m^{-1}$ depending on whether the snow is isotropic or anisotropic in the vertical direction, respectively.

Comparing parameterizations of $k_{eff}^{P}$ and $k_{eff}^{FEM}$, seen in Fig. 5, allows us to analyze which parameterizations represent the simulated $k_{eff}^{FEM}$ most accurately. Figure 5 shows that the majority of parameterizations appear to underestimate the $k_{eff}$ of samples with high anisotropy. Despite this, $k_{eff}^{Cal20}$ has a very similar $r^2$ value (0.96) to the polynomial fit of this dataset (Mac(II), $r^2 = 0.97$). Both $P =$ Yen and $P =$ Stm overestimate $k_{eff}$ when in the higher range of density values due to the adjustable coefficients in the original work

being optimized only in specific density ranges. When introducing an anisotropy parameter, $P =$ Löwe is well suited for low densities. However, similarly to $P =$ Yen and Stm, the $r^2$ value when measured for the entire dataset is heavily altered because this parameterization was not optimized for snow with a density above $550\,kg\,m^{-3}$. For this reason, we analyze higher and lower density ranges separately below.

### 4.1.1 Snow (density 50–550 $kg\,m^{-3}$)

Figure 5 shows that $k_{eff}^{Yen}$ and $k_{eff}^{Cal20}$ align on the $1:1$ line at low effective thermal conductivity values; this is also given in the relatively low (0.07) mean absolute error (MAE) in Table 2 for $P =$ Yen and Cal20. In contrast, $P =$ Stm has the highest MAE (0.15) for the lower density range, as it appears to overestimate $k_{eff}$ in most sub-samples, also seen in Fig. 5. $P =$ Mac(I) had the lowest MAE for any parameter which did not include anisotropy.

After introducing anisotropy into the parameterizations ($P =$ Löwe and Mac(II)), the MAE value reduces to 0.03 for the sub-samples in the lower density range. This indicated that anisotropy is critical for accurate $k_{eff}$ approximations.

### 4.1.2 Interfacial and icy layers (density > 550 $kg\,m^{-3}$)

In the upper range of $k_{eff}$ values, there is an underestimation when $P =$ Stm, Yen, and Löwe when compared to $k_{eff}^{FEM}$,

resulting in the large MAE values of 0.32, 0.31, and 4.40, respectively. However, this was expected, as Sturm et al. (1997), Yen (1981), and Löwe et al. (2013), as previously explained, did not include samples in the higher density range in their study. $P = \mathrm{Mac(I)}$, Cal20, and Mac(II) performed the best with the lowest MAE scores (0.15, 0.15, and 0.06, respectively), as these parameters were constructed for the complete range of density values. We corrected the $P = \mathrm{L\ddot{o}we}$ parameterization for higher densities (outlined in Sect. 2.2.1), resulting in the $k_{\mathrm{eff}}^{\mathrm{Mac(II)}}$ parameterization with an $r^2$ of 0.99 and low MAE values for both density ranges.

We have introduced two new thermal conductivity parameterizations; see Eqs. (4) and (5). The latter requires an anisotropy factor, which can, for now, only be measured in the laboratory using $\mu$CT. Using the SMP snow grain classification methods introduced by Kaltenborn et al. (2023) and an approximation of anisotropy for each grain type could be a future development but is beyond the scope of this study. Currently, we recommend using Eq. (4) or Calonne et al. (2019) when measuring the thermal conductivity of snow on Arctic sea ice when only density measurements or approximations are available. $k_{\mathrm{eff}}^{\mathrm{Mac(I)}}$ is used throughout this study when there were no co-measured anisotropy values, for example, when using the SMP. It is important to mention that calculations of $k_{\mathrm{eff}}^{\mathrm{FEM}}$ exclude convection, which would increase the thermal conductivity values. However, the convection is negligible compared to the conduction through the ice.

Anisotropy is critical for reducing uncertainty in calculations of snow's thermal conductivity. However, measuring anisotropy in the field is challenging and, as a result, limits our ability to conduct large-scale spatiotemporal studies without installing a $\mu$CT or shipping snow samples to a suitable laboratory for analysis. To this end, we propose a future study using techniques used in Kaltenborn et al. (2023), which classified snow grain type using the snow micro penetrometer force signal. By classifying the snow grain types and assigning a "typical" anisotropy to the classification, we have the ability to use a single instrument to obtain profiles of density, anisotropy, and consequentially thermal conductivity. If this method is successful, we can easily measure and upscale measurements of snow thermal conductivity throughout the cryosphere. This proposed method would introduce alternative uncertainties, such as misclassification of snow grain type and uncertainty in the anisotropy value assigned to a snow grain type, which would need addressing and evaluating in a follow-up study.

## 4.2 Spatial heterogeneity

Due to the high heterogeneity of the snow cover on Arctic sea ice, we used 3266 vertical snow profiles to estimate the thermal conductivity of the snow. These profiles were measured using the SMP after analyzing a suitable parameterization from the $\mu$CT dataset. As previously mentioned, the SMP does not have simultaneous anisotropy measurements,

so the parameterization $k_{\mathrm{eff}}^{\mathrm{Mac(I)}}$ was used. The SMP dataset consisted of 3266 profiles taken during this study period. This is highly representative of the landscape due to the spatial scale of the measurements taken over various conditions and a large measurement sample size.

This is the first time we have grouped a thermal conductivity measurement dataset by underlying ice type (FYI, SYI, and refrozen leads) and topographic feature (ridges) for one winter period. This has allowed us to analyze different features of importance for heat transfer. Figure 6 highlights that snow depth is highly dependent on the ridging of the ice, as known from other studies on sea ice ridging (Warren et al., 1999; Gradinger et al., 2010; Hames et al., 2022). $\rho^{\mathrm{SMP}}$ is slightly higher for refrozen leads, likely due to brine inclusions in the snow on refrozen leads during formation, which lowers the freezing temperature and increases the density. The same is for $\rho^{\mathrm{SMP}}$ measured at ridged sites, likely due to wind densification. However, the standard deviation is large enough for these variations not to be significant. $\overline{k_{\mathrm{eff}}^{\mathrm{Mac(I)}}}$ is derived from the $\rho^{\mathrm{SMP}}$; therefore, we see similar dependencies in the groups, as explained above.

The average $k_{\mathrm{eff}}^{\mathrm{FEM}}$ for all sub-samples of this dataset had a value of $0.27 \pm 0.17\,\mathrm{W\,K^{-1}\,m^{-1}}$, and the 1623 SMP profiles harmonically averaged between January and March profiles of $\overline{k_{\mathrm{eff}}^{\mathrm{Mac(I)}}}$ had an average value of $0.25 \pm 0.05\,\mathrm{W\,K^{-1}\,m^{-1}}$, seen in Fig. 4. The harmonic mean reduces the importance of extreme values in the sample. As a result, the $\overline{k_{\mathrm{eff}}^{\mathrm{Mac(I)}}}$ dataset has a smaller range in the histogram in Fig. 4. Despite the reduction in the range, the median value of $\overline{k_{\mathrm{eff}}^{\mathrm{Mac(I)}}}$ aligns with the median value of $k_{\mathrm{eff}}^{\mathrm{FEM}}$.

Including spatial heterogeneity in models is critical for improving heat transfer through the snow cover. Figure 4 compares the range of values of $\overline{k_{\mathrm{eff}}^{\mathrm{Mac(I)}}}$ and $\overline{k_{\mathrm{eff}}^{\mathrm{FEM}}}$ to the constant average value of $\overline{k_{\mathrm{eff}}^{\mathrm{Cal20}}}$ and $\overline{k_{\mathrm{eff}}^{\mathrm{Models}}}$ (also represented as $k_s$) which is equal to $0.30$–$0.33\,\mathrm{W\,K^{-1}\,m^{-1}}$ proposed by Maykut and Untersteiner (1971) and Semtner (1976). This snow thermal conductivity value is inferred from thermodynamic ice growth and is widely used in the modeling community (Sturm et al., 2002a; Lecomte et al., 2013; Holland et al., 2021). The breakdown of $\overline{k_{\mathrm{eff}}^{\mathrm{Mac(I)}}}$ for each ice type can be seen in Table 3. We propose that large-scale sea ice models test a lower average $k_s$ value of $0.25 \pm 0.05\,\mathrm{W\,K^{-1}\,m^{-1}}$ for snow on sea ice. We have calculated this using independent methods. We need to answer the following question: what would happen in Arctic sea ice models if the established value of $k_{\mathrm{eff}}$ was too high?

We conducted tests to determine the relationship between underlying ice and the thermal resistance of the snow. Nicolaus et al. (2009) identified a dependence of thermal conductivity depending on the underlying ice age. However, this is not the case for this dataset. By grouping thermal conductivity measurements by underlying ice type, we can conclude

that the thermal resistance is influenced by the HS (snow height) and less by the underlying ice type.

We found that the snowpack's thermal resistance $R$ on sea ice heavily depends on the ice surface topography as a result of different snow depths. Ridged areas showed approximately 3 times the thermal resistance compared to level ice areas. SYI and FYI areas have similar $R$ medians, with SYI areas having more significant heterogeneity than FYI areas. Finally, refrozen leads have the lowest $R$ and have a significant standard deviation. Sampling difficulties are likely one reason for these large standard deviations (especially on ridged and lead areas). Refrozen leads cannot be measured until there is sufficient ice thickness to walk on. However, different ages and seasons produce highly varying conditions on the leads (Clemens-Sewall et al., 2022), and our sampling was not focused on measuring different ages of refrozen leads throughout the season. This means that our sampling was likely not representative of the many conditions of refrozen leads and cannot be used to draw concrete conclusions about snow thickness and thermal resistance. The high variability in the ridge's $R$ values is due to the uneven snow distribution. SMP measurements were taken adjacent and perpendicular to the ridges to try and capture this heterogeneity.

## 4.3  Temporal change

The time component of this study shows that HS is highly variable, but the monthly median of SYI and lead areas remains consistent throughout the season. These ice-type categories were defined in situ using observations, and any saline snow areas were categorized as above FYI. Snow depth on FYI increased until March and shows a decrease after. This decrease in snow depth is likely due to the significant wind speeds during the storm event described by Wagner et al. (2022). This storm event could also have caused the increase in surface snow density in March, shown in Fig. 8. HS in ridged areas is highly heterogeneous and is likely due to the blocks within the ridges causing an uneven sea ice topography, causing high heterogeneity in snow accumulation. Temporal variability of the ridged sites could also be due to the operator selecting different ridge areas to measure or the sudden inaccessibility of different snowpit sites due to ice dynamics.

As HS is directly influencing $R$, we see no seasonal trend in $R$ values on level ice, with a value of $R = 515 \pm 404\,\mathrm{m^2\,K\,W^{-1}}$ on first-year ice and $660 \pm 475\,\mathrm{m^2\,K\,W^{-1}}$ on second-year ice. Therefore, we can conclude that the calculated values of $R$ remain consistent during winter but include high spatial heterogeneity due to snow depth variability. Ridged areas show a high heterogeneity throughout the season but no significant change in the average $R$ from January to April.

## 5  Conclusions

Using measurements of snow microstructure on different ice types and topographic features on Arctic sea ice for a 6-month winter period in the high Arctic, we have built upon previous work analyzing the seasonal evolution of snow's thermal conductivity (Sturm et al., 2002a; Calonne et al., 2019) using a method that has not previously been used on snow on sea ice. By evaluating the seasonal evolution and spatial heterogeneity of the snow's thermal conductivity and thermal resistance, we assessed the current thermal conductivity parameterizations and their performance for the range of possible snow densities. We present two new parameterizations, with and without anisotropy. We have explained that all scatter of thermal conductivity is related to the structural properties of density and anisotropy. Currently, the range of possible thermal conductivities associated with a single snow density is large enough to drastically influence sea ice growth model outputs (Lecomte et al., 2013). Therefore, we argue that anisotropy is a critical parameter for thermal conductivity parameterizations. Density is reasonably quick and efficient to measure in the field. However, we lack a method to obtain anisotropy in the field without transporting a $\mu$CT. One suggestion is to use the methods given in Kaltenborn et al. (2023) to identify the snow grain type and assign an anisotropy for each. This method would introduce uncertainties but allows for conducting more precise thermal conductivity measurements using the SMP alone.

Field measurements highlighted the need for a high sampling density to represent the spatial heterogeneity of thermal conductivity due to snow's high heterogeneity in the Arctic sea ice system. We conclude that the SMP dataset used in this study can be used to measure the thermal conductivity's heterogeneity, as it had a large sampling size over a wide variety of conditions. However, we believe that the community will benefit from future studies comparing different instruments and independent datasets from the MOSAiC expedition, which each measure the thermal conductivity of snow in the Arctic. In addition, we propose testing lower values of snow thermal conductivities in large-scale sea ice models. The average of $\overline{k_{\mathrm{eff}}^{\mathrm{Mac(I)}}}$ for all SMP winter measurements was $0.25 \pm 0.05\,\mathrm{W\,K^{-1}\,m^{-1}}$ for snow on sea ice. This indicates that $0.32 \pm 0.01\,\mathrm{W\,K^{-1}\,m^{-1}}$, currently used in sea ice modeling (Lecomte et al., 2013), may largely overestimate thermal conductivity. We also provide a breakdown of snow's thermal conductivity values per ice type and found that the averages ranged from 0.22 to $0.26\,\mathrm{W\,K^{-1}\,m^{-1}}$ (the overview can be seen in Table 3).

Due to the low correlation between thermal conductivity and ice type, we can confidently state that snow resistance is mainly influenced by snow height. We found an approximately 3 times higher thermal resistance on ridges $(1411 \pm 910\,\mathrm{m^2\,K\,W^{-1}})$, with extremely high spatial heterogeneity due to snow depth compared to level sea ice. The

thermal resistance of snow on level sea ice remains approximately constant, with a value of $R = 515 \pm 404\,\mathrm{m^2\,K\,W^{-1}}$ on first-year ice and $660 \pm 475\,\mathrm{m^2\,K\,W^{-1}}$ on second-year ice. We conclude that ridged and level areas must be treated separately in modeling, as thermal resistance is almost 3 times higher in ridged areas. High spatial heterogeneity of thermal resistance is apparent, but temporal changes in the snow cover are challenging to identify and interpret due to the highly dynamic and heterogeneous landscape.

## Appendix A: Geometric and thermal anisotropy

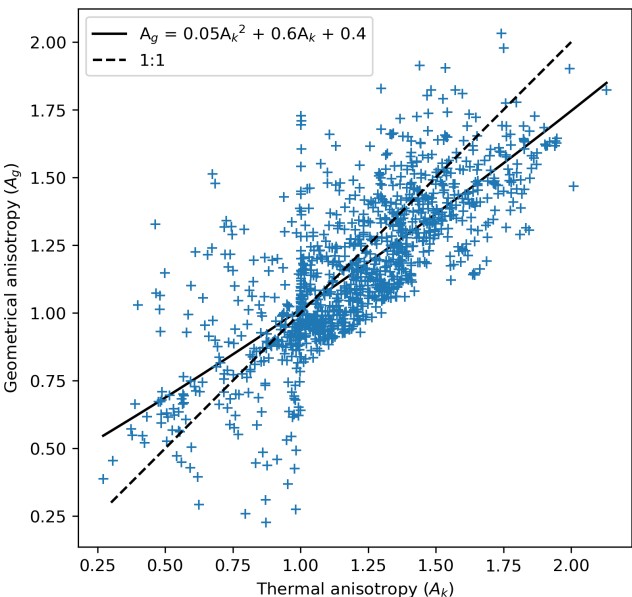

**Figure A1.** The comparison of geometrical and thermal anisotropy for each sub-sample of the MOSAiC snow on sea ice dataset. The polynomial fit is given in the legend.

## Appendix B: Anisotropy-based parameterization

To obtain a parameterization for the thermal conductivity that is applicable to the entire density range, we essentially start from Löwe et al. (2013), who were using a linear transformation of the so-called lower-bound $k_z^{(L)}$ to predict the FEM values. The lower bound is a known function $k_z^{(L)}(\phi, A_g)$ in terms of the ice volume fraction $\phi$ (related to the density via $\rho = \phi \rho_{ice}$) and the geometrical anisotropy $A_g$, which are known parameters from the tomography analysis. The function $k_z^{(L)}(\phi, A_g)$ is explicitly given in Löwe et al. (2013) in Eq. (2). However, a linear transformation of the bound cannot work for the entire density range, as detailed in Sundu et al. (2023) for the effective elasticity tensor. To this end, we use the same non-linear transformation proposed in Eqs. (11)

and (12) in Sundu et al. (2023) and propose

$$k_{eff}^{Mac(II)} = k_0 + k_{ice}\left(\frac{X^\beta}{\Omega(1-X) + X^{(\beta-1)}},\right) \tag{B1}$$

with $X = k_z^{(L)}/k_{ice}$ as a suitable parametric fit function with three fit parameters of $\Omega$, $\beta$, and $k_0$ that must be obtained by minimizing the differences between Eq. (B1) and the FEM estimates. The idea of the non-linear transformation of $k_z^{(L)}$ in Eq. (B1) is to capture the crossover between low densities (where the conductivity increases super linearly as reflected by the quadratic forms such as Eq. 4) and high densities (where the effective conductivity of snow must linearly approach the conductivity of ice).

*Data availability.* All snow datasets used in this article are published in Pangaea. The snowpit raw data are publicly available from https://doi.org/10.1594/PANGAEA.935934 (Macfarlane et al., 2021a). This dataset includes the SMP, $\mu$CT, density cutter, and SWE datasets.

Shortwave radiation measurements were obtained from the Atmospheric Radiation Measurement (ARM) user facility, a U.S. Department of Energy (DOE) Office of Science user facility managed by the Biological and Environmental Research Program, and are publicly available in the ARM data archive (https://doi.org/10.5439/1608608; Riihimaki, 2021).

Near-surface meteorology (2 m air temperature and wind speed) and surface energy flux measurements from the University of Colorado/NOAA surface flux team are available through the Arctic Data Center (https://doi.org/10.18739/A2VM42Z5F; Cox et al., 2021). TS2

*Author contributions.* ARM: data curation, investigation, visualization, formal analysis, and writing (original draft, review, and editing). HL: conceptualization, formal analysis, and writing (review and editing). LG: visualization, formal analysis, and writing (review and editing). DNW: data curation, investigation, and writing (review and editing). RD: Data curation, investigation, and writing (review and editing). RO: Formal analysis and writing (review and editing). MS: Project administration, funding acquisition, conceptualization, methodology, investigation, and writing (review and editing).

*Competing interests.* The contact author has declared that none of the authors has any competing interests.

*Acknowledgements.* The datasets used in this paper were produced as part of the international Multidisciplinary Drifting Observatory for the Study of Arctic Climate (MOSAiC), with the tag MO-SAiC20192020 and the project ID AWI_PS122_00. We thank all people involved in the expedition of the Research Vessel *Polarstern* (Knust, 2017) during MOSAiC in 2019–2020, as listed in Nixdorf et al. (2021). We want to thank SCANCO Medical AG for lending and supporting the use of the $\mu$CT 90 throughout the MOSAiC expedition.

*Financial support.* This research has been supported by the WSL-Institut für Schnee- und Lawinenforschung SLF (grant no. WSL_201812N1678), the Swiss Polar Institute (grant no. DIRCR-2018-003), and Horizon 2020 (grant no. 730965). TS3

*Review statement.* This paper was edited by Ruth Mottram and reviewed by two anonymous referees.

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

**Remarks from the language copy-editor**

CE1    Please clarify whether the slash should be replaced with "and", "or", or "and/or" in this instance.

**Remarks from the typesetter**

TS1    Due to the requested changes, we have to forward your requests to the handling editor for approval. To explain the corrections needed to the editor, please send me the reason why these corrections are necessary in a separate document together with the updated figure. Please note that the status of your paper will be changed to "Post-review adjustments" until the editor has made their decision. We will keep you informed via email.

TS2    You have defined the following data sets as assets to your paper: 10.1594/PANGAEA.952794. Please provide a statement on how your underlying research data can be accessed and ensure that all these data sets are properly cited as individual contributions with reference list entries including creators, title, and date of last access.

TS3    Please note that there is a discrepancy between funding information provided by you in the acknowledgements and the funding information you indicated during manuscript registration, which we used to create this section. Please double-check your acknowledgements to see whether repeated information can be removed from the acknowledgements or changed accordingly. If further funders should be added to this section, please provide the funder names and the grant numbers. Thanks.

TS4    Please check "Britannica, T.: Editors of encyclopaedia". (Author/Editor?: Title?)

TS5    Please provide persistent identifier (ISBN or DOI preferred).

TS6    Please provide page range or article number.

TS7    Please provide page range or article number.

TS8    Please provide page range or article number.

TS9    Please provide page range or article number.

TS10    Please provide page range or article number.

TS11    Please provide page range or article number.

TS12    Please provide page range or article number.