# Peer review of "Thermal Conductivity of Snow on Arctic Sea Ice"

_EGUsphere, 2023_

## Referee Comment (RC2)

**Review of Thermal conductivity of snow on Arctic Sea ice**

This manuscript describes the variation of the thermal conductivity of snow, its density and how it varies according to the type of ice it falls on based on the MOSAIC campaign. Snow on sea ice is still one of the more uncertain topics in the Arctic environment and with its large impact on the development of sea ice studies that improves the understanding of this are welcome. The MOSAIC campaign includes a large and valuable dataset, which has the potential of improving the understanding of the Arctic environment. That being said snow is difficult and the findings are accompanied with large uncertainties, which should be discussed more. In addition, some of the definitions are a bit loose. This is especially the case in the introduction. At last it would generalize the conclusions if additional data set were used or if parts of the dataset were used for calibration of the polynomial and other parts of the data set were used for validation.

I find that the manuscript needs some revisions and it should be read through and updated where language seems a bit rough as this makes it difficult to follow. This blurs the conclusion.

**Major**

Leads are normally openings of the sea ice due to dynamics. What the authors mean is likely refrozen leads where new, thin ice is formed and snow has just started to accumulate. This needs to be specified more clearly.

An increase in density until March can be seen, however the uncertainties are large enough. Is it statically significant?

Sometimes it seems as if snow and ice are mixed up. This is a bit confusing.

**General**

Figure text should be the same same font all the way

Figures with multiple sub figures would benefit from nation such as figure XXa, XXb….

**Minor**

line 6. This is related to the definition of leads. I don't see that leads are in the ice age category, thus the parenthesis is a bit misleading. I would rephrase this.

Line 7 I would change dynamics to seasonal variation or something like that.  I don't think that dynamics is the right word.

Line 10: I would not call the thermal resistance constant when uncertainties for first and second year ice is that high.

Line 12:  I think that the uncertainties should be mentioned here as well as it seems, as there are snow on sea ice. The uncertainties are large and the thermal resistance of snow on ridges can be the same as the two other categories.

Line 13: I would skip the last sentence that starts with: "The implications of our findings…." It does not really fit into an abstract.

Line 27-32 This describes heat transfer through snow, however ice Is mentioned. I am not sure that it is intended.

Line 35 remove "the". FEM is a general model not one specific.

Line 38 a model do not measure it calculates.

Line 41: I think that short fall should be shortcoming.

Line 44 please rephrase without parenthesis

Line 46. Rephrase as "on a snow"

Line 47 remove "," and replace with and

Line 52 A faster method? Faster in what way?

Line 65 I would rephrase line 65 to we can draw new conclusions about the…

Line 68: Where does the +- 0.01 originate from. I assume that this is a constant.

Line 72 coordinates are listed as N/S. North and South. One of them should be E/W

Line 83. I did not think that the instruments are the focus of this study. It should be the snow properties.

Line 96 replace "to" with "in order to"

Line 96 replace "," with and

Line 110: it is not clear what kair and kice are used for.

Line 114 to 120: This section is unclear. There are more x values than a, b, c. A table might help.

Line 125: Ag? Is it assumed that the reader reads Löwe (2013) as well?

Line 129. Is SMP and SMP force the same?

Line 132: Not sure what the sentence that starts and end in this line refer to.

Line 145: What is the current literature?

Line 168 It is not clear what the first sentence mean. Is it only high density variability or also KFEM and Ak?

Line 170 to 171. The reference to KFEM seems to be inserted into a discussion on density. I think that this should be reorganized.

Line 180: cover all. I would rephrase to a wide range.

Line 185. I assume equation 4. The parenthesis before Adapting due not have a start

Line 195: Is the seasonal variability significant compared to the uncertainty and variability of the measurements?

Line 208 Results in table 1 is mentioned but not really used.

Line 234 rephrase to avoid parenthesis

Line 255 and ridges? This should be rephrased to its own sentence.

Line 263: Any considerations on how the change of thermal conductivity would change the result of the models?

Line 338: This article that the conductivity of snow is 0.32 in sea ice modelling. Is this Crocus and SNOWPACK? I don't think that any of these models are normally included in sea ice modeling.

Figure 1

Remove " We could simulate...using the FEM method

Is the snow depth of each sample known?

Figure 7. Is there a dependency on the snow depth?

---

## Author Comment (AC1)

**RC1:**

**General comments**

The paper provides a dataset of effective thermal conductivity of snow on sea ice from tomographic images as well as from SnowMicroPen measurements. Data from the tomographic images are used to study the anisotropy of the thermal conductivity and assess existing parameterizations based on density. A new fitted regression is suggested. Data from the SMP measurements are used to study how thermal conductivity evolves with time and how it varies for 4 situations: snow lying on first year ice, on second year ice, on leads and snow on ridges. Associated variability in snow height and snow density are also presented and used to provide possible explanations for thermal conductivity variations.

The presented study will be largely beneficial for the snow and ice community as it provides a key dataset for the energy balance of sea ice. The dataset, involving in situ microCT, is unique. An impressive amount of field work and data analysis was provided and should be acknowledge. The idea to use SMP measurements to access more data and investigate spatial and temporal variations is new and seems well-suited.

However, the paper does not meet scientific standards in terms of data description and analysis. Some data lack thorough description or seem to be interpreted only half-way. Some statements are unclear or lack quantitative support. Additional work will be required to rigorously address the topic and provide a comprehensive presentation. I strongly encourage the authors to improve the paper as the topic can make for an important publication.

Thank you for this valuable feedback on the manuscript. I have implemented all your suggestions and I believe this has drastically improved the flow of the paper and its thoroughness. Your suggestions to conduct more statistical tests, improve the figures and make my explanations have suggested have been extremely beneficial throughout this review process.

The major issues regard

**- FEM computations of effective thermal conductivity**

It is unclear how the effective thermal conductivity was computed from the 3D images (see specific comment below) and which value of thermal conductivity is plotted in the different figures. Especially, in Figure 2, I believe it is the z-component of the thermal conductivity that is plotted, unlike what is mentioned. This point impacts the interpretation of the figure.

Thank you for this comment, We agree that the description of the FEM method is too short and some abbreviations could lead to confusion. We have made the following changes:

On line 108 we have included an explanation of the effective thermal conductivity ($k_{eff,z}$) alongside the definition. $K_{eff}$ is always the vertical component of the thermal conductivity tensor (z-direction). We shortened $K_{eff,z}$ to $K_{eff}$ throughout this manuscript.

Additionally, to avoid confusion $K_{eff}^X$ has now been changed to $K_{eff}^P$ where the P represents different parameterisations.

I have included an explanation of FEM in the section μ-CT to explain how the full 3-D tensors of effective thermal conductivity were calculated. In addition I have explained how we take the vertical temperature gradient (z-direction) and discard any lateral heat flow. Later I refer to the x, y, and z components when talking about calculations of thermal anisotropy.

We have added a figure to better explain the methods.

[Figure]

*Figure 1. a) A snow micro penetrometer (SMP) force signal showing stratigraphy of the snowpack during Event PS122-3_35-56. b) Three-dimensional reconstructions of two μ-CT samples showing a typical surface (top) and snow-sea ice interface (bottom) sample. c) The overview photo of the snowpit during Event PS122-3_35-56.*

Little information is provided on the relationship between thermal conductivity and microstructure. It is restricted to density although the authors have data of correlation length and geometrical anisotropy but they are not presented.

To provide more information on the relationship between these parameters we have taken the following steps: $A_K$, $A_g$ and correlation length values are published alongside the Pangaea dataset. We use the correlation lengths to obtain geometrical anisotropy, see Equation 1 and a figure has been added to Appendix 1 which shows the relationship between $A_K$ and $A_g$. We would also light to highlight the relationship between $A_K$ and thermal conductivity shown in Figure 3.

The new dataset needs to be compared to existing datasets for snow on sea ic*e.* The comparisons done with the Yen, Sturm and Calonne parameterizations are relevant, but in a first step it is indispensable to compare with the available data of thermal conductivity of snow on sea ice.

We have included an improved literature review in the introduction about the current thermal conductivity measurements in the Arctic. This has helped when re-writing parts of the discussion. Sturm paper is the only

existing dataset measuring thermal conductivity directly using the needle probe. We have investigated this method and compared the derived parameterization from Sturm. We reference the methods used by Merkourayadi and Hunke and discuss the values used for thermal conductivity in the modelling community further in the introduction.

We agree that the community will benefit from future studies comparing different instruments and independent datasets from the MOSAiC expedition which each measure the thermal conductivity of snow in the Arctic.

**- parameterizations of thermal conductivity**

The equations of the tested parameterizations, their domain of validity (density range) and information about how they were obtained (measurements, simulations) are lacking.

We have included a table with this information and expanded the section 4.1 Assessing existing parameterizations to two sub sections: Snow (density 50 – 550 kg m$^{-3}$) and Interfacial and icy layers (density > 550 kg m$^{-3}$) to improve the analysis of each range. In this study we have tested the complete range of snow and ice. These threshold labels are included in Figure 3 as vertical red lines which are further explained in the caption.

Up to 550 kg/m$^3$ was used for snow as we found some hard wind-packed depth hoar layers with this density value. Above 830 kg/m3 was classed as ice. The values in between we have assumed originate from icy layers in the snowpack and dense layers at the snow-ice interface. We were fortunate to have the complete range in the mosaic measurements. Thresholds were used to separate the statistical tests, of which a new table has been added.

The evaluation of the parameterizations lacks scientific rigor and is only qualitative (e.g. performances described only by "work well" (line 232)). The comparisons should be supported by statistical scores. Performance scores for low/high density and low/high anisotropy should be provided if discussed. The authors recommend using Equation (3) for snow on sea ice but arguments are elusive. The authors should review their statistical measure they use. $R^2$ values are insufficient for the conclusion they make.

I have added additional statistical analysis for snow density ranges (50-550 kg/m$^3$) and firn & ice density ranges (> 550 kg/m$^3$) under the sub-section "Assessing existing parameterizations". This includes an additional table giving the mean absolute error for individual cases.

The description of the proposed parameterization with anisotropy is incomplete and its purpose is unclear. First, the complete set of equations should be provided including the term k(L) from Löwe et al. 2013 as well as the values for the involved parameters Ω, β, k0. There are two components kz and kx described in Equation (2) of Löwe et al. 2013 and it is not mentioned which one is used. With this complete description, it is still unclear how this parameterization with anisotropy can be applied to other density datasets to derive the thermal conductivity. The benefit for the paper and the general profit are unclear.

Added details into an appendix and re-worded to explain that, instead of using the linear empirical adaptation from Loewe (2013), we used a similar empirical adjustment (Equations ~11 and 12) as suggested in Sundu (2023) for the elasticity tensor to cover the entire range of densities. More details are given in Appendix 2.

**The introduction**

lacks a comprehensive description of the state of the art. Some relevant studies are provided in the introduction but it needs reorganization and completion, so the picture of the state of the art and the current limitations become clear. The reader should know which measurements were done on the thermal conductivity for snow on sea ice, which tools were used, and what range of values was found. To give an example, Sturm et al. 2002 "Thermal conductivity and heat transfer through the snow on the ice of the Beaufort Sea" appears but there is no description. This comment also refers to the parameterizations and the modeling – how is currently model snow on sea ice heat transfer? In the current state, the introduction mixes studies for ice, snow and snow on ice, which is confusing.

Thank you for this suggestion, we have adapted the introduction to include more literature review of the current measurements of thermal conductivity of snow on Arctic sea ice and the approaches used in the modeling community.

**Specific comments**

27: It could be helpful to include one sentence of the main characteristics of snow on sea ice. This way the reader can follow when the importance of the different heat transport processes are discussed.

Added a sentence on the highly variable snow cover, explaining that variability stems in the meter-scale.

28 "3/ vapour diffusion between the snow grains" → do you refer to phase change?

Yes thank you for this comment. I have added phase change

34 "X-ray micro-computed tomography (microCT) has enabled snow research to advance by measuring the exact ice skeleton without damaging it (Riche and Schneebeli, 2010)" → this reference is not the paper introducing CT on snow.

Added reference to: Coleou, C., Lesaffre, B., Brzoska, J.B., Ludwig, W. and Boller, E., 2001. Three-dimensional snow images by X-ray microtomography. *Annals of glaciology*, *32*, pp.75-81.

39: "Density is currently used to parametrize thermal conductivity because it is a simple, low cost and quick measurement in the field" → no; first of all, it is because of the first order dependency between thermal conductivity and density.

Changed the sentence to explain how density is used to parametrise thermal conductivity because of the first-order dependency between thermal conductivity and density. Additionally, it is a simple, low cost and quick measurement in the field

51 "Spatial heterogeneity of the snow on sea ice requires a very high number of measurements, which can not only be realized by microCT." → replace with "The study of spatial heterogeneity of the snow on sea ice requires a very high number of measurements, which can not only be realized by microCT." Also, choose your wording between heterogeneity or variability, throughout the paper.

Thank you for this suggestion. I changed this throughout the manuscript and now use heterogeneity to explain landscapes and variability over a temporal scale.

62: "We up-scaled individual microCT" and throughout the paper → it is unclear what up-scaling refers to.

Added a sentence on how we related individual point measurements to a larger area by increasing the sample size

91: the cylindrical drill was operated by hand or was an electric drill used?

Added electric

151: The interest of computing thermal resistance R, compared to solely thermal conductivty, should be mentioned here.

If snow is considered as an interface between the atmosphere and the ice in models, it requires a thermal resistance rather than a conductivity. We have included a more detailed explanation in the test and added an extract from Bigdeli et al. (2020) which provides a very useful analogy.

Figure 2: It is unclear why snow samples with vertical anisotropy show higher values of $k_{eff}^{FEM}$ and inversely for horizontal anisotropy. Is $k_{eff}^{FEM}$ the average of $k_{eff}(x)$, $k_{eff}(y)$ and $k_{eff}(z)$, with $k_{eff}(x)$ the effective thermal conductivity computed under a heat flux in the x-direction and so on for y and z? This needs to be clearly defined in Section 2.2.1.

The more the structure is orientated in the direction of the temperature gradient the better the conduction even when the density remains the same. We take the z component of $k_{eff}$ throughout this study. $K_{eff}^{x}$ has now been changed to $K_{eff}^{P}$ where the P represents different parameterisations. We have included this in 2.2.1, thank you for the suggestion.

Figure 2: a zoom for the 0 – 300 density range would be helpful to read the data.

Added a zoomed area to the figure

107 : « The thermal conductivity of the micro-CT sub-samples … were compared ... as seen in Fig2.. » Figure 2 should be placed in Section 3.1 where it is actually described.

Moved the figure to be in section 2.2.1.

159 SWE is not defined

Thank you for noticing this. I have added the definition for SWE in section "Density Profiles"

161 « Due to a reduction of density... » → rewrite the sentence

This section of the manuscript has been adapted as we did not have enough data points to make this conclusion about the density reduction. We therefore do not cover this in the discussion as heavily.

175 : It would be informative to describe which type of snow has low $A_k$ values and high values (depth hoar?), if possible. To my knowledge, there has been little report of Ak values as low as 0.25 in the past, so it would be interesting to know what kind of snow it is.

We were able to measure icy layers due to the electric drill, I have added: "Extreme values of anisotropy in the lower range show icy layers and high values are depth hoar samples."

In addition, the range of values of the geometrical anisotropy should be provided here, so the link between both anisotropy ratios can be done.

Both Ak and Ag will be published alongside the dataset in Pangaea. I have added the below figure into the appendix

[Figure]

Legend of Figure 3 : description of the bar plots of color yellow and green is missing the line colours

Figure adapted

Section 3.1. Overall, the link between the thermal conductivity and snow microstructure could be better addressed. An idea could be to provide a CT image of a typical stratigraphy of snow on sea ice together with vertical profiles of density, correlation length, Ag, thermal conductivity, and Ak.

Added a figure of the snow collection protocol with a microCT figure and parameter extraction from the SMP.

185: remove parenthesis

Parenthesis removed

Legend Figure 4. Check the structure of the sentence.

This figure caption has been re-written according to the new labels in the plot.

187: "Without including anisotropy in the parameterization, kMac(I) eff is the best representation of keff, as it has the highest r2 value compared to this dataset". Only commenting on the performance of the regression that was fitted to the dataset is weak in terms of finding – it was not necessary to compare with other (independent)

parameterizations to come to this conclusion. To allow for fair comparisons, the validity range for the regression of Yen and Sturm, which have an upper density limit, should be provided.

Further statistical tests have been conducted in the section "Assessing existing parameterizations".

Section 3.3: provide a general description of Figure 5 before giving detailed comments.

Thank you for this suggestion. I have changed this to "Fig. \ref{fig:BarChart_Keff} shows the snow heights, snow density (measured using the SMP and the \cite{king2020local} parameterization) thermal conductivity and thermal resistance for each ice type and for ridge areas. This can be seen in the grey box plots in the background of Fig. \ref{fig:BarChart_Keff}."

188: "We use this parametrization and introduce the SMP to upscale our measurements of keff , of which we do not have corresponding Ak or Ag measurements". Is the second part of the sentence necessary? I don't understand the link with Ak and Ag here.

The meaning of this sentence is that iIf we did have Ag or Ak we wouldn't use this parameterisation we would be able to use Mac(II). I have added "Anisotropy is critical for reducing uncertainty in thermal conductivity, this is mentioned again in the discussion, and future work is suggested." I then discuss this further in the discussion.

190 to 194: This paragraph could be placed in Section 3.3, as it is about using SMP and harmonic means to explore spatial and temporal variability and not about comparing parameterizations.

Moved this paragraph, thank you for the suggestion

Figure 5: The legend needs to be reviewed (repetitions, incorrect wording). The meaning of the grey box and of the grey stars are missing.

I have removed the repeating legends and included the grey box and grey star meaning in the figure caption "These profiles are grouped by underlying ice type, topographic feature (seen in the grey bar charts in the background of the figure, with grey stars indicating the outliers), and month (seen in the coloured bar charts, of which the outliers are not shown"

197: As a general comment, the paper is lacking descriptions of figures or tables. For example, it is too not sufficient to write "Table 1 gives the median and standard deviation of each." without any supporting comment. A short sentence as "On average more snow is found on Ridges with HS = 335 mm and less on Leads with 84 mm" is very helpful for the reader.

Thank you for this suggestion, I have implemented it in the manuscript

Section 3.3 (related to the previous comment). The writing of this section should be improved. The description of the figures should include more quantitative descriptions. No value is provided in the entire section. For example, line 196 – 199: snow height trends are described using "increase" / "decrease" / "highly variable" / "high spatial heterogeneity" without providing supporting values.

Added supporting values of ranges and re-worded the text

199 "Leads and ridges show consistently high spatial heterogeneity" → Is it shown on Figure 5?

Changed to "The range of snow depth on ridges (0 to > 1000 mm) shows consistently high spatial heterogeneity throughout the winter season; therefore, temporal changes are less discernible than in FYI and SYI areas."

Figure 6. Provide a comprehensive description of the figure. Increase the figure resolution as the text (n=, M =) is difficult to read on a printed copy of the paper. Some sentences of the legend are actually figure analysis and should be placed in the text.

Changes made to the figure

208 "keff has a slightly lower median on FYI and SYI, compared to leads and ridges with a higher keff". The first comment should rather be that values of harmonical mean of keff for each site are very close to each other. The grouping FYI-SYI and leads-ridges is not clear.

Changed this paragraph to $\overline{k_{\mathrm{eff}}^{\mathrm{Mac(I)}}}$ has a standard deviation between 0.04 and 0.06 W K$^{-1}$ m$^{-1}$ for all ice types the difference between the median $\overline{k_{\mathrm{eff}}^{\mathrm{Mac(I)}}}$ of these ice types is 0.04 W K$^{-1}$ m$^{-1}$, meaning the values are very close and are not significantly different. This data can be found in Table \ref{tab:overview}.

The temporal increase in density and $k_{\mathrm{eff}}$ is rather until between February and March depending on the site (Fig 5).

Changes made in section 3.3, due to insufficient data to draw conclusions about the density decrease we have decided to reduce the emphasis on this in our discussion.

217 "When using these parametrizations to investigate the heterogeneity of the snow cover, the microCT was not an ideal method for obtaining a representative sampling of the snow cover due to the time required for one measurement" → suggestion: To investigate the spatial variability of the snow cover, the microCT is not an ideal method due to the time required for one measurement

Thank you for the suggestion

229 "A snow sub-sample with a density of 400 kg m−3 can have a thermal conductivity value ranging from 0.2 W K−1m−1 to 0.6 W K−1m−1 if the snow is isotropic or anisotropic, respectively." → This should read as "if the snow is isotropic or anisotropic in the vertical direction, respectively". Anisotropy in the horizontal direction leads to even lower values, if the interpretation of this figure is correct.

Added "in the vertical direction" as we only assess thermal conductivity in the z-direction. I have made adjustments to the manuscript to clarify this.

Section 4.2: It could be interesting to discuss the possible impacts of the ice type and the topography on snow and so on the thermal conductivity. What was the initial motivation to study snow on different ice types, did you expect differences? (this could be included in the introduction).

The initial motivation is that this is the first time we have data to be able to do this. Thermistor strings and point measurements have not allowed this in past studies. Added "This is the first time we have been able to group a dataset of thermal conductivity measurements by underlying ice type (FYI, SYI and leads) and topographic feature (ridges) for one winter period. This has allowed us to analyse different features of importance for heat transfer."

Line 259 – 266: this paragraph is not about spatial variability as it is about comparing measured values with values used in models, so it appears out of subject here.

Re-phrased this paragraph as the spread of values in figure 3 is due to spatial heterogeneity of the slow and I have now explained that one value in the models is related to this in Figure 3.

265: It is an open question what the influence of convection is, but we need to answer the question…" → in the introduction, it is mentioned that convection is reduced in wind slabs on sea ice. More context is required to understand why we refer to convection here.

Removed this sentence to read "We need to answer the question…"

292 "As snow undergoes metamorphism, we expect its thermal conductivity to increase as the density increases." → Not all types of metamorphism involve a density increase; temperature gradient metamorphism can keep snow at about constant density. Explanation needs improvement.

Removed this sentence, I agree this was incorrect.

293 "We now work to understand the process causing a reduction in density after March." → provide order of magnitude of the reduction in density. Are we trying to understand a gap of 5 or 100 kg/m3?

We have found that there are not enough measurements in May to draw and concrete statistical analysis. As a result we have excluded May and mention this in the discussion.

298 "Fresh snowfall as input would lower the average density. A layer with low thermal conductivity kMac(I)eff leads to a drastic decrease in the average thermal conductivity. We can see in Wagner et al. (2022) that we had fresh snowfall during this period." → Be more specific that "we had snowfall". It could be mentioned that this is not seen in the snow height data in Figure 5, as there is no increase. Also, snowfalls are always fresh.

Removed this section from the discussion, see the above comment.

312 "Penetration of the hard density layers at the snow-ice interface became thinner due to sublimation" →not understandable.

This has been removed and the discussion has been restructured accordingly.

318 "Crocus and SNOWPACK simulate lower layers with high density and high thermal conductivity, and surface layers with low values for both variables (Domine et al., 2019), whereas we have presented the opposite." → It is not shown in the paper that you have opposite results. There is no clear trend in the vertical profile of thermal conductivity, as seen in Figure 1, and it is not described in the text. No density profiles are shown. So additional data / figure should be provided to support this comment or this paragraph should be deleted. Also, SNOWPACK and Crocus are not defined.

I have removed this paragraph

321 "Due to the combination of this seasonal trend in density and kMac(I)eff snow depths, we see no change of R" → wording

Re-worded this sentence

340 "It was found that a combination of fresh snowfall, high wind speeds causing erosion and re-deposition (initiating a SWE reduction during a storm event, as shown in Wagner et al. (2022)), vapour diffusion within the snow and changes at the snow-ice interface (further analysis of this interface is needed to draw concrete conclusions. However, this is not in the scope of this study), could all result in the density reduction across the snow profile." → this sentence should be shortened.

Thank you for the suggestion the sentence is now shortened

Conclusion: The conclusion could be improved to provide a clearer picture of the main contributions / findings of the paper.

Introduction, discussion and conclusions have been adapted according to the changes and suggestions above. Thank you for your help improving this manuscript.

---

## Author Comment (AC2)

**RC2**

Review of Thermal conductivity of snow on Arctic Sea ice

This manuscript describes the variation of the thermal conductivity of snow, its density and how it varies according to the type of ice it falls on based on the MOSAIC campaign. Snow on sea ice is still one of the more uncertain topics in the Arctic environment and with its large impact on the development of sea ice studies that improves the understanding of this are welcome. The MOSAIC campaign includes a large and valuable dataset, which has the potential of improving the understanding of the Arctic environment. That being said snow is difficult and the findings are accompanied with large uncertainties, which should be discussed more. In addition, some of the definitions are a bit loose. This is especially the case in the introduction. At last it would generalize the conclusions if additional data set were used or if parts of the dataset were used for calibration of the polynomial and other parts of the data set were used for validation.

Thank you for your valuable feedback on the manuscript. I have implemented all your suggestions and have made substantial changes to the introduction, discussion and conclusion upon receiving these comments and I believe the paper has improved substantially as a result of your help and comments.

I find that the manuscript needs some revisions and it should be read through and updated where language seems a bit rough as this makes it difficult to follow. This blurs the conclusion.

**Major**

Leads are normally openings of the sea ice due to dynamics. What the authors mean is likely refrozen leads where new, thin ice is formed and snow has just started to accumulate. This needs to be specified more clearly.

Thank you for this suggestion, I have changed this throughout the manuscript

An increase in density until March can be seen, however the uncertainties are large enough. Is it statically significant? Sometimes it seems as if snow and ice are mixed up. This is a bit confusing.

We concluded that there was insufficient data to draw any conclusions about May, so we have excluded this from the manuscript. I have explained further in the manuscript when snow and ice are being analysed, I agree due to the nature of the dataset we have data for the complete range of snow - ice, however this isn't used consistently throughout the manuscript as sometimes we use thresholds, so I agree it was previously difficult to follow. Hopefully this is now clarified throughout the manuscript.

**General**

Figure text should be the same same font all the way

I believe the same font is used, does this comment refer to the size? Please specify which figure you are referring to If I have not made sufficient changes to the re-submitted version.

Figures with multiple sub figures would benefit from nation such as figure XXa, XXb....

Thank you for this suggestion, I have added this to all sub-figures.

**Minor**

line 6. This is related to the definition of leads. I don't see that leads are in the ice age category, thus the parenthesis is a bit misleading. I would rephrase this.

This is now changed to refrozen leads

Line 7 I would change dynamics to seasonal variation or something like that. I don't think that dynamics is the right word.

Thank you for the suggestion, changed dynamics to seasonal variation

Line 10: I would not call the thermal resistance constant when uncertainties for first and second year ice is that high.

Changed to "the average thermal resistance of snow on level sea ice remains approximately constant with substantial variability with values of…"

Line 12: I think that the uncertainties should be mentioned here as well as it seems, as there are snow on sea ice. The uncertainties are large and the thermal resistance of snow on ridges can be the same as the two other categories.

Added "with substantial spatial variability"

Line 13: I would skip the last sentence that starts with: "The implications of our findings…." It does not really fit into an abstract.

Thank you for this suggestion, I have removed the last sentence

Line 27-32 This describes heat transfer through snow, however ice Is mentioned. I am not sure that it is intended.

Changed to "Snow's thermal conductivity and insulating properties directly impact heat transfer from the underlying sea ice to the atmosphere and directly inhibit ice growth in the winter season."

Line 35 remove "the". FEM is a general model not one specific.

Changed this throughout the manuscript, thank you for the suggestion

Line 38 a model do not measure it calculates.

Changed measure to calculate

Line 41: I think that short fall should be shortcoming.

Changed shortfalls to shortcomings

Line 44 please rephrase without parenthesis

rephrased

Line 46. Rephrase as "on a snow"

Rephrased to "realised the influence of temperature on the thermal conductivity."

Line 47 remove "," and replace with and

Reworded to "\cite{calonne2019thermal} created upper bounds to ensure that the thermal conductivity is in agreement with the thermal conductivity of ice at specific temperatures in the higher density ranges."

Line 52 A faster method? Faster in what way?

Added: A faster method is needed (the $\mu$-CT on MOSAiC took 7 hours to measure 10 cm of snow).

Line 65 I would rephrase

Removed previously impossible

line 65 to we can draw new conclusions about the…

thank you for the suggestion

Line 68: Where does the +- 0.01 originate from. I assume that this is a constant.

This originates from the different values used in the models, change this to "value of 0.31 to 0.33"

Line 72 coordinates are listed as N/S. North and South. One of them should be E/W

Thank you for noticing this, I made a mistake with the coordinate system, which has now been changed.

Line 83. I did not think that the instruments are the focus of this study. It should be the snow properties.

Changed to "we analyse the MOSAiC snowpit"

Line 96 replace "to" with "in order to"

Thank you, changes made to the manuscript

Line 96 replace "," with and

Change made to the manuscript

Line 110: it is not clear what kair and kice are used for.

This is the thermal conductivity of ice and air, which are the basis for the calculations of the thermal conductivity of snow,

Line 114 to 120: This section is unclear. There are more x values than a, b, c. A table might help.

Provided an overview table

Line 125: Ag? Is it assumed that the reader reads Löwe (2013) as well?

Ag is reference on line 96.

Line 129. Is SMP and SMP force the same?

The SMP is the instrument, the SMP force is the output. Rephrased this to "The snow micro penetrometer (SMP) instrument measures the penetration force resistance of a snow profile"

Line 132: Not sure what the sentence that starts and end in this line refer to.

Added "(more details of the measurement protocol can be found alongside the published dataset and datapaper \cite{macfarlane2021smp})."

Line 145: What is the current literature?

Changed this to "After testing the listed parameterizations in Table…"

Line 168 It is not clear what the first sentence mean. Is it only high density variability or also KFEM and Ak?

Re-worded this sentence

Line 170 to 171. The reference to KFEM seems to be inserted into a discussion on density. I think that this should be reorganized.

Reworded this to include density and Ak variability and the impact of this on KFEM values.

Line 180: cover all. I would rephrase to a wide range.

The ranges are now included in the text

Line 185. I assume equation 4. The parenthesis before Adapting due not have a start

Thank you for noticing this

Line 195: Is the seasonal variability significant compared to the uncertainty and variability of the measurements?

Included more details in this section about the averages and ranges of the data.

Line 208 Results in table 1 is mentioned but not really used.

I agree, I have now used a lot more data in the main text

Line 234 rephrase to avoid parenthesis

Removed parenthesis

Line 255 and ridges? This should be rephrased to its own sentence.

Thank you for this suggestion

Line 263: Any considerations on how the change of thermal conductivity would change the result of the models?

In general a lower thermal conductivity would increase ice growth in the winter season due to the snow cover being less thermally insulating, Merkouriadi (2017) is now referenced in the paper and they conduct a very interesting study on thermal conductivity parameterisation influence on ice growth.

Line 338: This article that the conductivity of snow is 0.32 in sea ice modelling. Is this Crocus and SNOWPACK? I don't think that any of these models are normally included in sea ice modeling.

Removed this section now as a result of previous suggestions, but thank you for this

Figure 1 Remove " We could simulate…using the FEM method Is the snow depth of each sample known?

Changed we could simulate to "we simulated". Snow depth is indicated in the figure through height from/to

Figure 7. Is there a dependency on the snow depth?

Due to the high variability within the snow profiles (seen in figure 1) there is no dependency of force on depth. We see a large influence of thermal resistance to snow depth, as mentioned in the previous sections.

---

## Author Response (AR1)

Dear Dr. Ruth Mottram and the two anonymous reviewers,

Thank you for your time and effort spent reviewing this manuscript. I have uploaded the revised manuscript with the suggested amendments.

Kind regards,

Amy Macfarlane